# Climate impact on the development of Pre-Classic Maya civilization

**Authors**
Kees Nooren[1], Wim Z. Hoek[1], Brian J. Dermody[1,2], Didier Galop[3], Sarah Metcalfe[4], Gerald Islebe[5] and Hans Middelkoop[1].

**Affiliations**
[1]Utrecht University, Faculty of Geosciences, 3508 TC Utrecht, The Netherlands;
[2]Utrecht University, Copernicus Institute of Sustainable Development, 3508 TC Utrecht, The Netherlands;
[3]Université Jean Jaurès, CNRS, UMR 5602 GEODE, 31058 Toulouse, France;
[4]University of Nottingham, School of Geography, Nottingham NG7 2RD, UK
[5]El Colegio de la Frontera Sur, Unidad Chetumal Herbario, Chetumal, AP 424 Quintana Roo, Mexico;

*Correspondence to*: Kees Nooren (k.nooren@gmail.com)

**Keywords**
Pre-Classic Maya period, Central Maya Lowlands, climate record, beach ridges, palaeo-precipitation, 500-yr periodicity, 2.8 ka event.

**Abstract**
The impact of climate change on the development and disintegration of Maya civilization has long been debated. The lack of agreement among existing palaeoclimatic records from the region has prevented a detailed understanding of regional-scale climatic variability, its climatic forcing mechanisms, and its impact on the ancient Maya. We present two new palaeo-precipitation records for the Central Maya Lowlands, spanning the Pre-Classic period (1800 BCE – 250 CE), a key epoch in the development of Maya civilization. A beach ridge elevation record from world's largest late Holocene beach ridge plain provides a regional picture, while Lake Tuspan's diatom record is indicative of precipitation changes at a local scale. We identify centennial-scale variability in palaeo-precipitation that significantly correlates with the North Atlantic $\delta^{14}C$ atmospheric record, with a comparable periodicity of approximately 500 years, indicating an important role of North Atlantic atmospheric-oceanic forcing on precipitation in the Central Maya Lowlands. Our results show that the Early Pre-Classic period was characterized by relatively dry conditions, shifting to wetter conditions during the Middle Pre-Classic period, around the well-known 850 BCE (2.8 ka) event. We propose that this wet period may have been unfavorable for agricultural intensification in the Central Maya Lowlands, explaining the relatively delayed development of Maya civilization in this area. A return to relatively drier conditions during the Late Pre-Classic period coincides with rapid agricultural intensification in the region and the establishment of major cities.

## 1. Introduction
During the last decades, a wealth of new data has been gathered to understand human-environmental interaction and the role of climate change in the development and disintegration of societies in the Maya Lowlands (e.g., Akers et al., 2016; Douglas et al., 2015, 2016; Dunning et al., 2012, 2015; Lentz et al., 2014; Turner and Sabloff, 2012). Previous studies have emphasized the impact of prolonged droughts and their possible link with social downturn, such as the Pre-Classic Abandonment and the Classic Maya Collapse (Ebert et al., 2017; Hoggarth et al., 2016; Lentz et al., 2014; Kennett et al., 2012; Medina-Elizalde et al., 2010, 2016; Haug et al., 2003; Hodell et al., 1995, 2001, 2005). Less attention has been given to episodes of excessive rain and floods that may also have severely impacted ancient Maya societies (e.g. Iannone et al., 2014). Evidence for such impacts is found in the fact that floods, as well as droughts, are an important theme depicted in the remaining ancient Maya codices (Fig. 1) (Thompson, 1972), and Maya mythological stories (Valásquez Garciá, 2006).

One of the main challenges in palaeoclimatic reconstructions is to unravel climate from human induced changes. Maya societies played a key role in the formation of the landscape, but the degree of human induced impact remains highly debated (Hansen, 2017; Beach et al., 2015; Ford and Nigh, 2015). For example, it is proposed that the increase in sedimentation rate after 1000 BCE at Lake Salpeten (Anselmetti et al., 2007) and Peten-Itza (Mueller et al., 2009) is related to human induced soil erosion. However, other high resolution lake records from the area do not show a significant increase in sedimentation rate during the Pre-Classic or Classic period (e.g. Wahl et al., 2014), and past volcanic activity could have been responsible for the deposition of smectite rich clay layers in inland lakes

(Tankersley et al., 2016; Nooren et al., 2017a). Palynological records from the Central Maya Lowlands (CML, Fig. 2) show no evidence of widespread land clearance and agriculture before ~400 BCE (Wahl et al., 2007; Islebe et al., 1996; Leyden et al., 1987), and there is growing consensus that the decline in the percentage of lowland tropical forest pollen during the Pre-Classic period (Galop et al., 2004; Islebe et al., 1996; Leyden et al, 1987) was caused by climatic drying instead of deforestation (Torrescano and Islebe, 2015; Wahl et al., 2014; Mueller et al., 2009).

In this paper, we present two new palaeo-precipitation records reflecting precipitation changes in the CML. The records span the Pre-Classic period (1800 BCE – 250 CE), when Maya societies in the CML transformed from predominantly mobile hunter-gatherers in the Early Pre-Classic period (e.g. Inomata et al., 2015; Coe, 2011; Lohse, 2010), to complex sedentary societies that founded impressive cities like El Mirador by the later part of the Pre-Classic period (Hansen, 2017; Inomata and Henderson, 2016). The period of rapid growth in these centralized societies probably occurred much later than previously thought, sometime after the start of the Late Pre-Classic period around 400 BCE (Inomata and Henderson, 2016). This raises the question for the reason behind the delayed development of societies in this area, which was to become the core area of Maya civilization during the following Classic period (250 – 900 CE). There is recent evidence that climate during the Middle Pre-Classic Period (1000 – 400 BCE) may have been less stable than recently reported (Ebert et al., 2017). We propose that anomalously wet conditions could have been unfavourable for the intensification of maize-based agriculture, which formed the underlying subsistence economy responsible for the development of many neighbouring Mesoamerican societies during this period.

The CML have been intensively studied, and several well-dated speleothem, palynological, and limnological records have been obtained for this area (Díaz et al., 2017; Akers et al., 2016; Douglas et al.; 2015; Wahl et al., 2014; Kennett et al., 2012; Mueller et al., 2009; Metcalfe et al., 2009; Domínguez-Vázquez and Islebe, 2008; Galop et al., 2004; Rosenmeier et al., 2002; Islebe et al., 1996) (Fig. 2 and A1). However, palaeo-precipitation signals from these records and those from adjacent areas in the Yucatan and Central Mexico exhibit large differences among records (Fig. A2), making the reconstruction and interpretation of larger-scale precipitation for the region a challenge (Lachniet et al., 2013, 2017; Douglas et al., 2016; Metcalfe et al., 2015). Existing climate reconstructions mostly represent local changes and are predominantly based on oxygen isotope variability, although some new proxies have been introduced recently (e.g. Díaz et al., 2017; Douglas et al., 2015).

We present a regional-scale palaeo-precipitation record for the CML, extracted from world's largest late Holocene beach ridge sequence at the Gulf of Mexico coast (Fig. 2B). The beach ridge record captures changes in river discharge resulting from precipitation patterns over the entire catchment of the Usumacinta River and thus represents regional changes in precipitation over the CML (Nooren et al., 2017b). Currently the annual discharge of the Usumacinta river is approximately 2000 $m^3$/s, corresponding to ~40 % of the excess or effective rain falling in the 70,700 $km^2$ large catchment (Nooren et al., 2017b). Mean annual precipitation within the catchment is ~2150 mm, with 80 % falling during the boreal summer, related to the North American or Mesoamerican Monsoon system (Lachniet et al., 2013, 2017; Metcalfe et al., 2015). The interpretation of the beach ridge record is supported by a new multi-proxy record from Lake Tuspan, an oligosaline lake situated within the CML. The lake receives most of its water from a relatively small catchment of 770 $km^2$ (Fig. 2) and hence provides a local precipitation record, to complement the regional signal from the beach ridge sequence.

*Regional palaeo-precipitation signal*
The coastal beach ridges consist of sandy material originating from the Grijalva and Usumacinta rivers, topped by wind-blown beach sand (Nooren et al., 2017b). Although multiple factors determine the final elevation of the beach ridges, it has been shown that during the period 1775 ± 95 BCE to 30 ± 95 CE (at 1σ), roughly coinciding with the Pre-Classic period, beach ridge elevation was primarily determined by the discharge of the Usumacinta river. Low elevation anomalies of the beach ridges occur in periods with increased river sediment discharge, which in turn is the product of high precipitation within the river catchment. Under these conditions, beach ridges develop relatively rapidly, and are exposed to wind for a shorter period. In contrast, during periods of drought, sediment supply to the coast is reduced, resulting in a decreased seaward progradation rate of the beach ridge plain. This leaves a longer period for aeolian accretion on the beach ridges near the former shoreline, resulting in higher beach ridges (Nooren et al., 2017b). Hence, variations in beach ridge elevation reflect changes in rainfall over the Usumacinta catchment, and thereby represent catchment-aggregated precipitation, rather than a local signal. The very high progradation rates and the very robust age-distance model

(Fig. A3), with uncertainties of the calibrated ages not exceeding 60–70 years (at 1σ), effectively allow
the reconstruction of palaeo-precipitation at centennial time scales.
*Local palaeo-precipitation signal: Lake Tuspan record*
During dry periods, a reduced riverine input of fresh water and a lowering of the lake level enhance the
effect of evaporation and increase the salinity of the lake water. Diatom communities within oligo- to
hypersaline lakes are strongly influenced by lake water salinity (Reed, 1998; Gasse et al., 1995), and
we therefore determined diatom assemblage changes within the Lake Tuspan sediment record (Fig. 3)
to reconstruct palaeo-salinities of the lake water, reflecting palaeo-precipitation in the lake's catchment
via changes in the balance of precipitation - evaporation.
**2. Methods**
*Beach ridge sequence*
Beach ridges elevations were extracted from a Digital Elevation Model (DEM) of the coastal plain
along the transects indicated in Fig. 2 (Nooren et al., 2017b). The DEM is based on LiDAR data
originally acquired in April-May 2008 and processed by Mexico's National Institute of Statistics and
Geography (INEGI), Mexico. The relative beach ridge elevation is defined as the difference between
the beach ridge elevation and the long-term (~500 yr) running mean (Fig. A3). The age distance model
is based on 35 AMS [14]C dated terrestrial macro-remains (mainly leaf fragments isolated from organic
debris layers), and 20 OSL dated sand samples (determined on small aliquots of quartz grains) (Nooren
et al., 2017b).
*Lake Tuspan*
Two parallel cores, Tuspan cores B and C, were taken with a Russian corer (type GYK) in shallow
water near the inflow of the Rio Dulce (not to be confused with the Rio Dulce that drains lake Izabal in
eastern Guatemala), not far from core A which has been studied for pollen (Galop et al., 2004). Semi-
quantitative analyses of Si, S, K, Ca, Ti, Mn and Fe were conducted on both cores with an X-ray
fluorescence core scanner (type AVAATECH) at 0.5 cm intervals. Deposits of large floods were
identified on the basis of elevated concentrations of Si, Fe, Ti and Al (Davies et al., 2015), with peak
concentrations exceeding a one-standard-deviation threshold above the mean.
Core C was investigated for amorphous silica, charred plant fragments, and diatoms (Fig. 3, A4 and
A5). The core was subsampled at 4-12 cm contiguous intervals, each interval representing 25-80 years.
In addition, 37 1-cm samples (representing ~6.5 yr) were processed using the method outlined by
Battarbee (1973) to determine diatom concentrations and short term variability (decadal scale).
Subsamples were treated with HCl (10 %) to remove calcium carbonate. Large organic particles were
removed by wet sieving (250 µm mesh), and charred plant fragments > 250 µm were counted under a
dissection microscope. Remaining organic material was removed by heavy liquid separation using a
sodiumpolywolframate solution with a density of 2.0 g/cm$^3$. A silicious residue, denoted 'amorphous
silica' was subsequently removed by heavy liquid separation using a sodiumpolywolframate solution
with a density of 2.3 g/cm$^3$, and dry weight was determined after drying of the samples at 105$^o$C.
Slides were prepared from the remaining material. Diatoms were identified, counted and reported as
percentages of the total diatom sum, excluding the small and often dominant *Denticula elegans* and
*Nitzschia amphibia* species. These species show a large variability on short time scales (Fig. A5), and
are not indicative for changes at centennial time scale. We relate changes in diatom assemblages
mainly to lake water salinity changes. The first principal component on the entire assemblage (PC-1) is
interpreted as a palaeosalinity indicator. Diatom taxonomy is mainly after Patrick and Reimer (1966;
1975) and Novelo, Tavera, and Ibarra (2007). We identified *Plagiotropis arizonica* following
Czarnecki and Blinn (1978), *Mastogloia calcarea* following Lee et al. (2014), and *Cyclotella*
*petenensis* following Paillès et al. (2018).
The age-depth model for core C is based on seven AMS radiocarbon dated terrestrial samples and
stratigraphical correlation with core A (Fleury et al., 2014). We used a linear regression between the
available radiocarbon dated samples (Fig. A7) which is comparable with the age-depth model of Fleury
et al. (2014) for the time window between ~2500 BCE and 1000 CE.
*Wavelet transfer functions*
The relation between our beach ridge and diatom record and other palaeo-precipitation records from
the Maya Lowlands and nearby regions (figure A1 and A2) were investigated by wavelet coherence
(CWT) analyses using the software developed by Grinsted et al. (2004). We also applied CWT to
compare our record with North Atlantic ice drift record (Bond et al., 2001) and the Northern
Hemispheric atmospheric $\delta^{14}$C record (Reimer et al., 2013) to gain an understanding of the forcing of
the regional changes in precipitation we observe. The record of drift ice from the North Atlantic is
bimodally distributed, oscillating between periods of low and high concentrations of hematite-stained
grains. The timeseries was therefore transformed into a record of percentiles based on its cumulative
distribution function to avoid leakage of the square wave into frequency bands outside the fundamental
period (Grinsted et al., 2004). CWT applies Monte Carlo methods to test for significance. In this case
we set the alpha value at 5%. Time periods and periodicities enclosed within the black lines of in our
wavelet analysis indicate common power between timeseries with 95% confidence.
**3. Results**
*Beach-ridge record*
As described above, when beach ridge elevation is largely driven by the discharge of the Usumacinta
River, periods of relative high (low) beach ridges correspond to relatively drier (wetter) conditions in
the Usumacinta catchment (Fig. 2). The beach-ridge record shows clear centennial scale variability,
with an exceptionally dry phase centred around 1000 ± 95 BCE, and a subsequent pronounced wet
phase centred around 800 ± 95 BCE.
*Diatom record Lake Tuspan*
The sediment stratigraphy of core C can be divided into two main units. Below 4.3 m the core is clearly
laminated, and 0.5-4-cm-thick dark palaeoflood-layers contrast with the predominantly light-coloured
calcareous deposits. The flood-layers are characterized by elevated detrital input, resulting in elevated
concentrations of Si, amorphous silica, and charred plant fragments. The average recurrence time of
large floods was approximately 50 years. Sediments from 0.25 to 4.3 m depth are vaguely laminated,
with three distinct dm-thick turbidite layers (Fig. 3).
The interpretation of PC-1 (Fig. 3) as an indicator of lake salinity and hence relative dryness (see
Methods) is supported by the fact that low/negative PC-1 values are driven by relatively high
percentages of *Plagiotropis arizonica* (Fig. 3), a diatom species characteristic of high-conductivity
water bodies (Czarnecki and Blinn, 1978) as well as other benthic, high salinity/alkalinity species such
as *Anomoeoneis sphaerophora* and *Craticula cuspidata* (Fig. A4).
A drastic change from dry to wet conditions occurred around 4.3 m depth (¬1100 BCE), with the loss
of salinity tolerant taxa and higher proportions of freshwater taxa such as *Eunotia* sp. and *Cyclotella*
*petenensis* (Fig. 3). This coincides with the observed lithological shift from clearly to vaguely
laminated sediments. After relatively high/positive PC-1 values during the Middle and Late Pre-Classic
Period we observe a decreasing trend in PC-1 values during the following Classic Period, indicating a
gradual increase in lake water salinity. Low PC-1 values between 800 – 950 CE are in accordance with
many palaeorecords from the area (Fig. A2) indicating periods of prolonged droughts during the Late
Classic Period.
*Wavelet transfer function*
Wavelet coherence (WTC) analysis (Grinsted et al., 2004) indicates in-phase coherence between the
beach ridge record and the recently extended and revised calcite $\delta^{18}$O speleothem record from Macal-
Chasm cave (Akers et al., 2016) (Fig. A7). The in-phase relationship between the two records is
significant above a 5% confidence level at centennial timescales during the Pre-Classic Period. We did
not find significant relationships between the beach ridge record and other palaeo-precipitation records
from the CML, nor with records from the Yucatan and Central Mexico (Fig. A2), except for a
significant in-phase coherence at a centennial time scale with the *Pyrgophorus* sp. $\delta^{18}$O record from
Lake Chichancanab (Hodell et al., 1995).
**4. Discussion**
**4.1 Climate change in the CML during the Pre-Classic period**

	*Early Pre-Classic period (1800 – 1000 BCE)*
Both beach ridge and diatom records indicate that the onset of the early Pre-Classic period was
relatively dry (Fig. 4). Despite the predominantly dry conditions, large floods still occurred, as
demonstrated by the repetitive input of fluvial material into Lake Tuspan. Periods with the highest
fluvial sediment input coincided with periods of increased input of charcoal into Lake Peten-Itza
(Schüpbach et al., 2015) (Fig. A2). Because the CML were still sparsely populated during the Early
Pre-Classic period (Inomata et al., 2015) we relate the presence of charcoal to the occurrence of
wildfires.
After a transition to wetter conditions between 1500 and 1400 BCE, we observe a drying trend that
culminated in a prolonged dry period at the end of the Early Pre-Classic period centred around $1000 \pm$
95 BCE. Although this exceptionally dry phase is less apparent from Lake Tuspan's diatom record
(Fig. 3), it has been recorded at many other sites within the CML. At Lake Puerto Arturo, high $\delta^{18}O$
values in the gastropod *Pyrgophorus* sp. indicate that this was the driest period since 6300 BCE (Wahl
et al., 2014), and the recently extended and improved speleothem $\delta^{18}O$ record from Macal Chasm
indicates that this dry period was probably at least as severe as any prolonged droughts during the
Classic and Post-Classic period (Akers et al., 2016). Dry conditions are reflected in high
Ca/$\Sigma$(Ti,Fe,Al) values at Lake Peten-Itza (Mueller et al., 2009), indicating elevated authigenic
carbonate ($CaCO_3$) precipitation relative to the input of fluvial detrital elements (Ti, Fe and Al) during
this period; water level at this large lake must have dropped by at least 7 m (Mueller et al., 2009).
*Middle Pre-Classic period (1000 – 400 BCE)*
Both the beach ridge and the Lake Tuspan diatom records indicate a change to wetter conditions
around 1000-850 BCE, causing major changes in hydrological conditions in the CML (Fig. 4). The
diatom assemblages in the Lake Tuspan record show a major change in composition. Species indicative
of meso- to polysaline water almost completely disappear, and are replaced by species indicating fresh
water conditions (Fig. 4). In the lake sediments, this transition is also marked by a lithological shift
from clearly to vaguely laminated sediments that lack repetitive large flood layers, while charred plant
fragments are almost absent until ~400 BCE. Similar abrupt lithological transitions were reported from
Lake Chichancanab (Hodell et al., 1995) and Lake Peten-Itza (Mueller et al., 2009), and Wahl et al.
(2014) describe a regime shift at Puerto Arturo. The sudden reduction in charred plant fragments
around ~1000 BCE at Lake Tuspan coincides with reduced concentrations of charcoal at Lake Peten-
Itza (Fig. A2) (Schupbach et al., 2015) and Laguna Tortuguero, Puerto Rico (Burney and Pigott
Burney, 1994) indicating rapid climatic changes over a large spatial scale.
*Late Pre-Classic period (400 BCE – 250 CE)*
Both beach ridge and diatom record (Fig. 4) indicate that a relatively dry period occurred by the onset
of the Late Pre-Classic period, which has not been identified in other proxy records from the region
(Fig. A2), although high *Pinus* pollen percentages in the pollen record from Petapilla pond near Copan
(McNeil, 2010) during this period may indicate dry conditions, as high *Pinus* pollen percentage at
highland sites could be indicative of drier conditions (Domínguez-Vázquez and Islebe, 2008). The
diatom record at Lake Tuspan (Fig. 3) shows a general increase in lake water salinity, indicating a
gradual shift to drier conditions in the Late Pre-Classic period.
**4.2 Precipitation variability**
*Precipitation variability over long time scales*
The observed general drying trend over the last few thousand years is probably related to the southward
shift of the ITCZ during the late Holocene. The shift occurred in response to orbitally-forced changes
in insolation (Haug et al., 2001), causing a gradual Northern Hemisphere cooling versus Southern
Hemisphere warming (Fig. 4), thereby shifting the ITCZ towards the warming southern hemisphere
(Schneider et al., 2014). Wetter conditions during the Middle Pre-Classic period may reflect a more
northerly position of the ITCZ, which may be related to stronger easterly tradewinds and the less
frequent occurrence of winter season cold fronts, as beach ridge morphological changes suggest
(Nooren et al., 2017b).
*Centennial-scale precipitation variability*
The coherence between the beach ridge record and the relatively well-dated Macal-Chasm speleothem
record gives us confidence that these records reflect regionally coherent variability at centennial
timescales during the Pre-Classic period. Interestingly, the beach ridge record is significantly in anti-

phase with the North Atlantic ice drift record (Bond et al., 2001) and the Northern Hemispheric atmospheric $\delta^{14}$C record during the Pre-Classic Period (Reimer et al., 2013) (Fig. 5), suggesting an important role of North Atlantic atmospheric-oceanic forcing on precipitation in the CML. The Northern Hemispheric atmospheric $\delta^{14}$C record shows a 512-yr periodicity (Stuiver and Braziunas, 1993), which is similar to the observed ~500 year periodicity of the beach ridge record during the Pre-Classic period. Such a centennial scale periodicity is not apparent in Lake Tuspan's diatom record (Fig. 3), nor in any of the other palaeo-precipitation records from the Maya Lowlands (Fig. A2), but has been identified in the Ti record from Lake Juanacatlán in the highlands of Central Mexico (Jones et al., 2015). This periodicity has been related to the intensity of the North Atlantic thermohaline circulation and variations in solar activity (Stuiver and Braziunas, 1993).

The coherence with fluctuations in solar irradiance is most evident during the 850 BCE (2.8 ka) event, related to the Homeric Grand Solar Minimum. At that time, a strong decrease in the total solar irradiance resulted in higher atmospheric $^{14}$C production and a change to cooler and wetter conditions in the Northern Hemisphere (e.g. Van Geel et al., 1996), and apparently also a shift to wetter conditions in the CML, evident from our two new palaeo-precipitation records (Fig. 4). This correlation should not be used as an analogue for modern precipitation variability, when periods of lower solar activity are associated with lower Usumacinta River discharge and hence less precipitation in the CML (Fig. A8). Probably due a more northerly mean position of the ITCZ during the Pre-Classic Period precipitation responded differently to solar forcing then today.

It has previously been suggested that there was a coherent response to the late Holocene southward shift of the ITCZ in both northern South America and the Maya Lowlands (Haug et al., 2003), implying that the beach ridge record should be in-phase with the Cariaco Ti record (Haug et al., 2001). Although the Cariaco record indicates large centennial-scale variability in precipitation over northern South America (Fig. 4), this variability is not significantly correlated with the beach-ridge record. The correlation improved slightly using an updated age-depth model for Cariaco (Fig. A9), but remains insignificant, probably as a consequence of uncertainties in the chronological control of both records or because of a more prominent influence of the Northern Atlantic climatic forcing mechanisms in the Maya Lowlands.

**4.3 Precipitation versus human development in the CML**
Our records indicate that the Early Pre-Classic period in the CML was relatively dry. During that period, the CML were still sparsely populated by moving hunter-gatherers. It is highly likely that before maize became sufficiently productive to sustain sedentism, the karstic lowlands were less attractive for humans than the coastal wetlands along the Gulf of Mexico and Pacific coast, where natural resources were abundantly present to successfully sustain a hunting/gathering subsistence system (Inomata et al., 2015). Reliance on cultivated crops, most notably maize, rapidly increased after the onset of the Middle Pre-Classic period around 1000 BCE (Rosenswig et al., 2015). Between 1000 and 850 BCE, under still dry conditions, there is evidence for increased maize agriculture in the Pacific flood basin (Rosenswig et al., 2015), and within the Olmec area on the Gulf of Mexico coast (Arnold III, 2009), and maize grains (AMS $^{14}$C dated to 875 ± 29 BCE) have been found as far as Ceibal within the CML (Inomata et al., 2015). We speculate that wetter conditions after 850 BCE might have been unfavourable for further development of intensive agriculture in the CML. This is supported by palynological evidence, indicating that widespread land clearance and agriculture activity did not occur before ~400 BCE (Wahl et al., 2007; Galop et al., 2004; Islebe et al., 1996; Leyden et al., 1987), despite some early local agricultural activity (Wahl et al., 2014; Rushton et al., 2013; McNeil et al., 2010). A return to drier conditions during the Late Pre-Classic period coincided with an expansion of maize-based agriculture in the CML, and communities within the Maya Lowlands show strong and steady development (Hansen, 2017; Inomata and Henderson, 2016). Hence, major development of Maya civilization in the Central Maya Lowlands occurred only after the onset of the Late Pre-Classic period, when climate became progressively drier, in line with earlier findings that drier conditions were favourable for agricultural development in the CML (Wahl et al., 2014). Changes in the distribution of rainfall probably also changed, and large floods, most evident during the Archaic and early Pre-Classic period, occurred much less frequently after approximately 1000 BCE.

**5. Conclusions**
For the first time a regional palaeo-precipitation record has been reconstructed for the Central Maya Lowland (CML), based on an exceptionally well dated high resolution beach ridge record. This record indicates centennial scale precipitation fluctuations during the Pre-Classic period that are not always

registered in local records, adding valuable new insights into larger scale climatic forcing mechanisms
for the CML. The generally poor correlation between the regional and local palaeo-precipitation
reconstructions are probably related to spatial precipitation variability, and chronological uncertainties
of many records. Additional research of beach ridge formation processes are needed to extend this
regional precipitation reconstruction to the Classic and Post-Classic period.
We have also generated a local scale palaeo-precipitation record using diatoms preserved in a core
from Lake Tuspan, thereby adding an alternative proxy to the relatively high number of local
reconstructions predominantly based on oxygen isotope variability. We recognise, however, that
diatom preservation is often poor in the carbonate lakes across the wider region. As a result, the
correlation between these two reconstructions is variable through time.
Although the occurrence of a prolonged drought during the end of the Early Pre-Classic period, which
we report here, is evident in other palaeo-precipitation reconstructions from the CML, the subsequent
wet period during the Middle Pre-Classic period, registered in both our new records, is less evident
elsewhere. Although many researchers have focused on the impact of drought on the development and
disintegration of Maya societies, one should consider this prolonged wet period as potentially
unfavourable for the development and intensification of agriculture in the CML, particularly in the
wetter areas.
Our results provide evidence that North Atlantic atmospheric-oceanic forcing plays an important role in
the modulation of the observed centennial scale precipitation variability, however further studies are
required which compare well-dated terrestrial reconstructions that capture regional signals with solar
and oceanic reconstructions to gain a better understanding of climate forcing mechanisms, both in the
CML and across the wider region.

**Acknowledgements**
This research is supported by the Netherlands Organization for Scientific Research (NWO-grant
821.01.007). The LiDAR data was generously provided by INEGI, Mexico. We acknowledge Philippe
Martinez, Jacques Giraudeau and Pierre Carbonel for the XRF core scan measurements, and we would
like to thank Peter Douglas, Pete Akers and Gerald Haug for providing their data. We thank Konrad
Hughen for valuable suggestions to update the age-depth model for Cariaco's sediment core 1002D.
We thank Ton Markus for improving the figures, and Mark Brenner and an anonymous reviewer for
their valuable comments to improve the paper.

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

**Figure captions**

Figure 1: The image on page 74 of the Codex Dresden depicts a torrential downpour probably associated with a destructive flood (Thompson, 1972).

Figure 2: A large part of the Central Maya Lowlands (outlined with a red dashed line) is drained by the Usumacinta (Us.) River (A). During the Pre-Classic period this river was the main supplier of sand contributing to the formation of the extensive beach ridge plain at the Gulf of Mexico coast (B).

Periods of low rainfall result in low river discharges and are associated with relatively elevated beach ridges. The extent of the watersheds of the Usumacinta and Dulce River is calculated from SRTM 1-arc data (USGS, 2009). Indicated are archaeological sites (squares) and proxy records discussed in the text; Tu= Lake Tuspan, Ch = Lake Chichancanab, PI = Lake Peten-Itza, MC = Macal Chasm Cave, and PA = Lago Puerto Arturo.

Figure 3: Summarized proxy record of Lake Tuspan sediment core C. In the lithological column black lines represent large flood layers and grey boxes turbidites. Ca and Si (in cps = counts per second) are presented here as % of total counts. Vertical lines (red) in the (amorphous) Si graphs indicate the one-standard-deviation threshold above the mean. For the diatom record only the relative abundance of 'key' diatom species are shown here. *Denticula elegans* and *Nitzschia amphibia* were excluded from the diatom sum. Notice abrupt change around 1100 BCE.

Figure 4: Comparison of the Lake Tuspan and beach ridge record (A) with local and proximal records from Macal-Chasm cave (Akers et al., 2016) and the Cariaco basin (Haug et al., 2001)(B). We used an updated age-depth model for the Cariaco record (Fig. A9). Climate records related to North Atlantic atmospheric-oceanic forcing are indicated in panel C, including the drift ice reconstruction from the North Atlantic (Bond et al., 2001), the Northern Hemispheric residual atmospheric $\delta^{14}$C content (Reimer et al., 2013), the Northern-to Southern hemispheric temperature anomaly (Schneider et al., 2014) and reconstructed Total Solar Irradiance (TSI) (Steinhilber et al., 2012).

Figure 5:Wavelet Transform Coherence (WTC) analysis between the beach ridge record and the Northern Hemispheric atmospheric $\delta^{14}$C record (Reimer et al., 2013)(A) and the North Atlantic ice drift record (Bond et al., 2001)(B). The beach ridge record is significantly in anti-phase with both records at approximately 500-yr time scale, indicating an important role of North Atlantic atmospheric-oceanic forcing on precipitation in the Maya Lowlands during the Pre-Classic period. The 5% significance level against red noise is shown as a thick contour. Arrows indicate phase difference, with in-phase relationship between records if arrows point to the right.

**Appendix: Additional figures**

Figure A1: Location of proxy records indicated in figure A2 and/or mentioned in the main text. A: Northern Maya Lowlands (Tz=Tzabnah, PL=Punta Laguna, RS=Rio Secreto, Ch=Chichancanab and Si=Silvituc), the Central and Southern Maya Lowlands (PA=Puerto Arturo, NRL=New River Lagoon, Tu=Tuspan, PI/Sa=Peten-Itza and Salpeten, MC/CH=Macal Chasm and Chen Ha, and YB=Yok Balum), the Maya Highlands (Oc/Na= Ocotalito and Naja, Am=Amatitlan, and Pet=Petapilla). B: Central Mexico (Jua=Juanacatlan, CdD=Cueva de Diablo, Jx=Juxtlahuacan, and Alj=Aljojuca) and the marine record from the Cariaco (C) basin. Annual precipitation (1950-2000) calculated with WorldClim version 1.4 (release3); Hijmans et al. (2005). Long-term (1958-1998) mean ITCZ position and wind at 925 hPa (m.s$^{-1}$) for July after Amador et al. (2006), based on NCED/NCAR Reanalysis data (Kalnay et al., 1996).

Figure A2a: Palaeoprecipitation records from the Central Maya Lowlands and Yucatan; Beach ridge elevation and Tuspan diatom record (this study), compiled record of Central Peten and Yucatan (Douglas et al., 2016), Salpeten and Chichancanab dD wax-corr. (Douglas et al., 2015), Salpeten $\delta^{18}$O (Rosenmeier et al., 2012), Peten-Itza $\delta^{18}$O (Curtis et al., 1998), Puerto Arturo $\delta^{18}$O (Wahl et al., 2014), Macal Chasm $\delta^{18}$O (Akers et al., 2016), Chen Ha $\delta^{18}$O (Pollock et al., 2016), Yok Balum $\delta^{18}$O (Kennett et al., 2012), Rio Secreto $\delta^{18}$O (Medina-Elizalde et al., 2016), Silvituc DV-pollen (Torrescano-Valle and Islebe, 2015), Chichancanab S and $\delta^{18}$O (Hodell et al., 1995), Punta Laguna $\delta^{18}$O (Hodell et al., 2007), and Tzabnah $\delta^{18}$O (Medina-Elizalde et al., 2010). Notice that the y-axis is sometimes reversed, so that excursions above the x-axis always indicate relatively drier conditions.

Figure A2b: Proxy records from the Central Maya Lowlands, the Maya Highlands and Central Mexico. Peten-Itza charcoal (Schüpbach et al., 2015), Peten-Itza pollen (Islebe et al., 1996), Amatitlan *Aulacoseira* and *Pinus* (Velez et al., 2011), Petapilla *Pinus* (McNeil et al., 2010), Naja *Pinus* (Domínguez-Vázquez and Islebe, 2008), Ocotalito Sr (Díaz et al., 2017), Aljojuca $\delta^{18}$O (Bhattacharya et al., 2015), Cueva del Diablo $\delta^{18}$O (Bernal et al., 2011), Juxtlahuaca $\delta^{18}$O (Lachniet et al., 2015, 2017), and Juanacatlan Ti 15-point running mean (Jones et al., 2015).

Figure A3: Age-distance model for beach ridge transect B. We refer to Nooren et al. (2017b) for further details.

Figure A4: Diatom record for lake Tuspan core C. Diatom concentrations (*1000 valves/g dw) were determined on 37 selected 1-cm samples and diatom percentages (only the 'key species' are shown here) were determined on the 123 subsamples at 4-12-cm contiguous intervals. The small and often dominant *Denticula elegans* and *Nitzschia amphibia* species were excluded from the diatom sum.

Figure A5: Detailed diatom record around one of the larger flood events ~1200 BCE.

Figure A6: Age-depth model for Tuspan core C. The age-depth model is based on a linear interpolation between calibrated ages of radiocarbon dated terrestrial macroremains from core A (Galop et al., 2004) and core C (Fleury et al., 2014). The model is most reliable for ages between ~2500 BCE and 1000 CE.

Figure A7: Wavelet Transform Coherence (WTC) analysis between the beach ridge record and the Macal Chasm $\delta^{18}$O record (Akers et al., 2016). The 5% significance level against red noise is shown as a thick contour. Arrows indicate phase difference, with in-phase relationship between records if arrows point to the right.

Figure A8: Mean annual discharge of the Usumacinta river at Boca del Cerro (Banco Nacional de Datos de Aguas Superficiales, consulted in January 2017) compared with the total solar irradiance (TSI). The TSI is comprised of the reconstruction from 1700-2004 (Krivovo at al., 2007), concatenated with observations from the Total Irradiance Monitor (TIM) on NASA's Solar Radiation and Climate Experiment (SORCE) from 2005-2011 (Kopp and Lean, 2011). 4.56 watts are added to the TIM measurements as previous reconstructions were calibrated against less accurate measuring equipment, compared with the TIM instrument, which led to an overestimation of TSI.

Figure A9: Updated age-depth model for Cariaco core 1002D. Original model (Haug et al., 2001) has been based on a linear interpolation of calibrated ages. We applied a 4th-order polynomal fit through modelled ages calculated with a P_sequence model (Oxcal 4.2) (Bronk Ramsey, 2009, 2016): k = 10, Marine13 calibration curve, delta R = 15 ± 50, one outlier: NSRL-13050.

Fig. 1

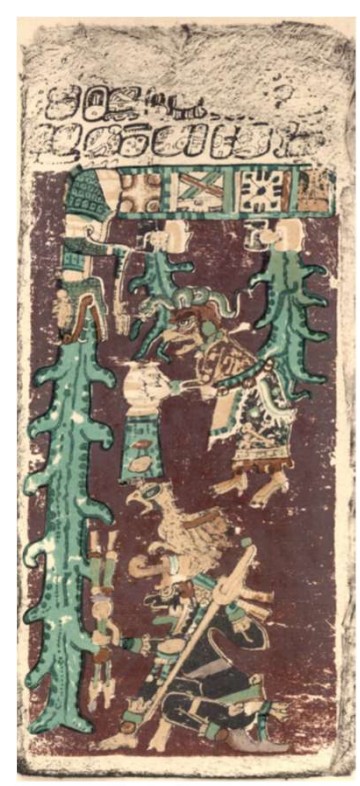

**C**

Year CE

rel. elevation (m)

| | | | | | | |
|---|---|---|---|---|---|---|
| -4500 | -3500 | -2500 | -1500 | -500 | 500 | 1500 |

**Archaic**          **Pre-Classic**          **Classic**  **Post-C.**

E    M    L          E  L  T    E  L

**B**

4300 B.C.E.

3400 B.C.E.

1800 B.C.E.

900 B.C.E.

150 C.E.

1050 C.E.

1550 C.E.

Gulf of Mexico

5 km

0

m+MSL
0

B2

**A**

○ ● cave / lake
■ arch. site
▲ volcano
▢ CML
🟥 beach-ridge plain
▨ catchment Us. R.
🔵 catchment Dulce R.

90°W

Gulf of Mexico

Chichen Itza

Ch

LOWLANDS

La Venta

El Mirador

Tu  PA  Tikal
     Pi/Sa    MC

Seibal

18°N

HIGHLANDS

Pacific

Copan

Cuauhtémoc

N

0    200 km

9247

Fig. 2

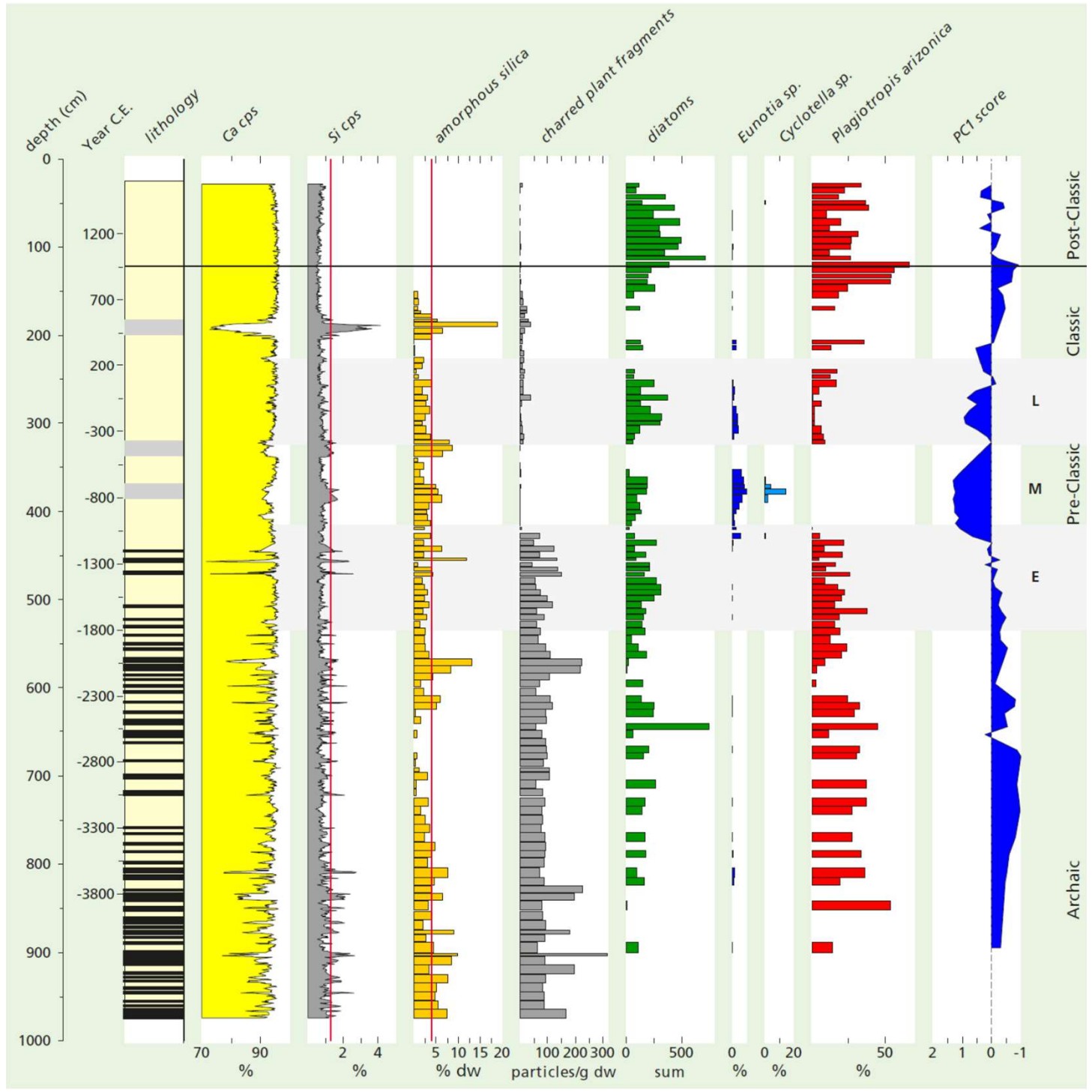

Fig. 3

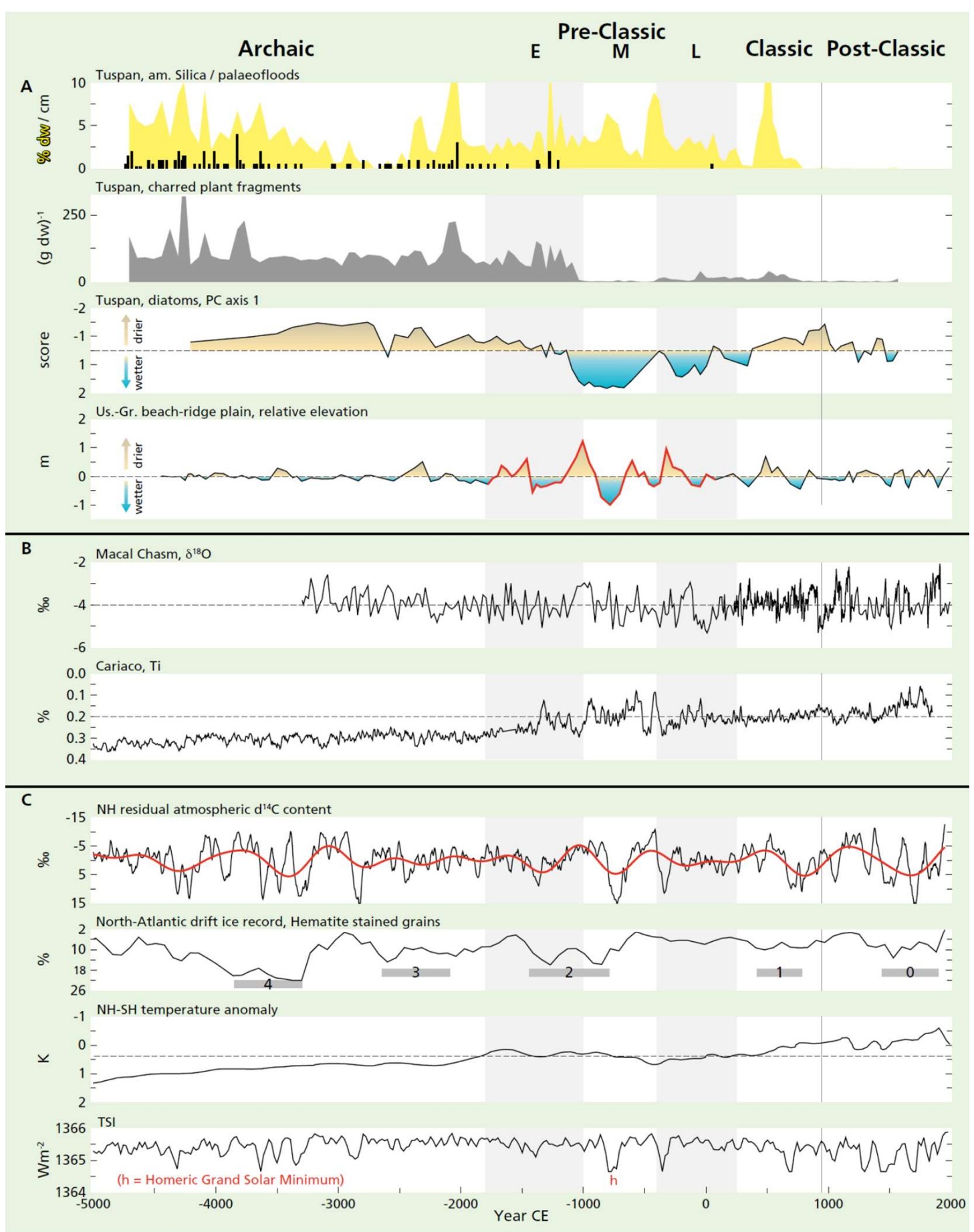

Fig. 4

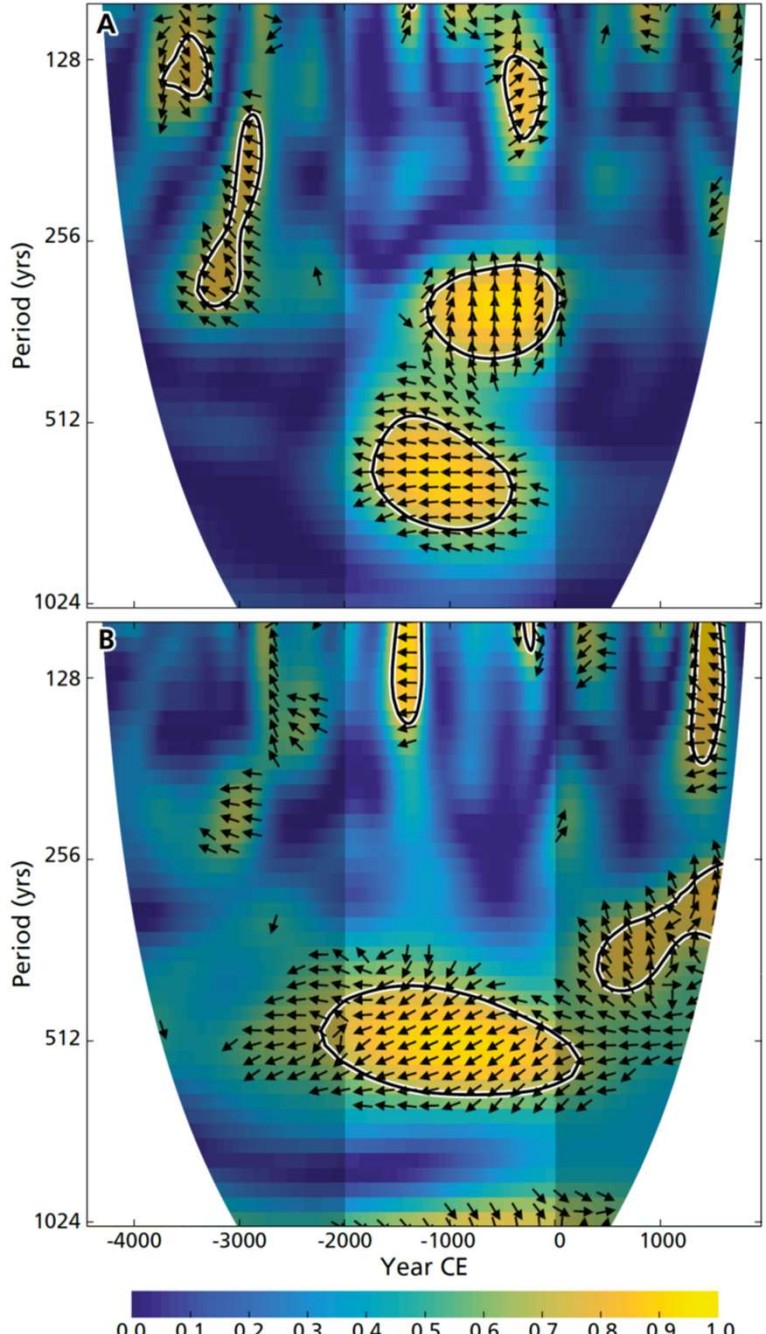

Fig. 5

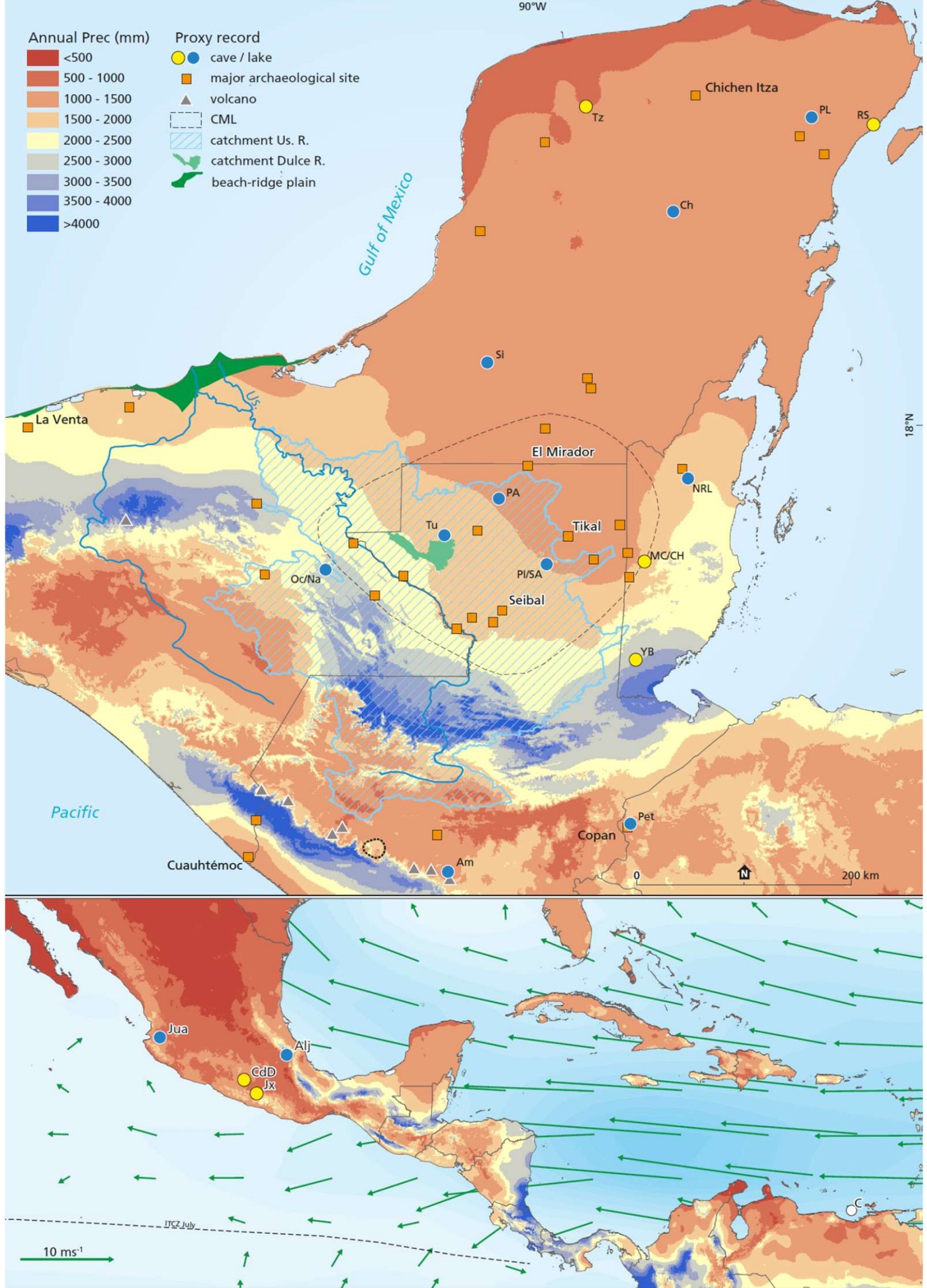

Fig. A1

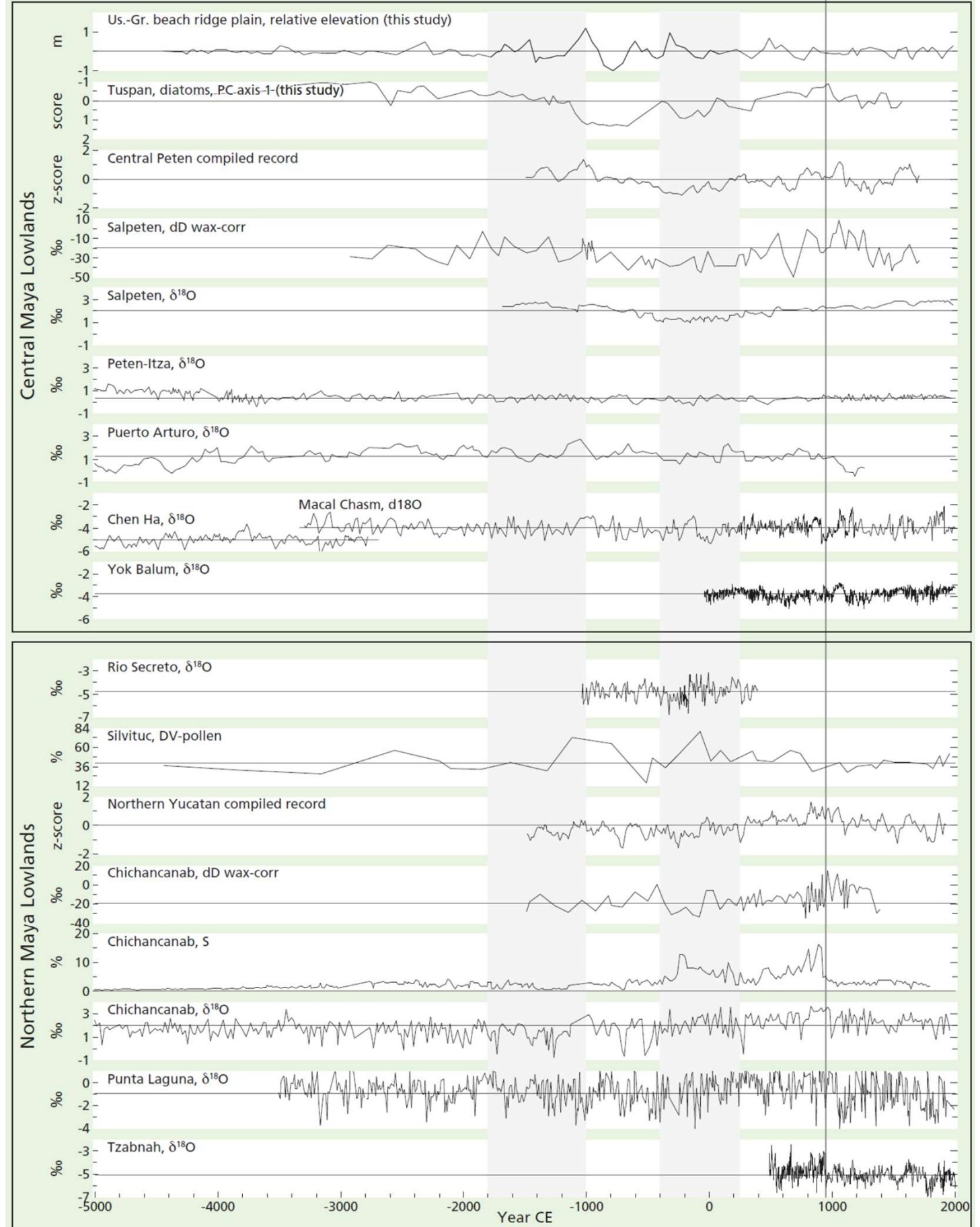

Fig. A2a

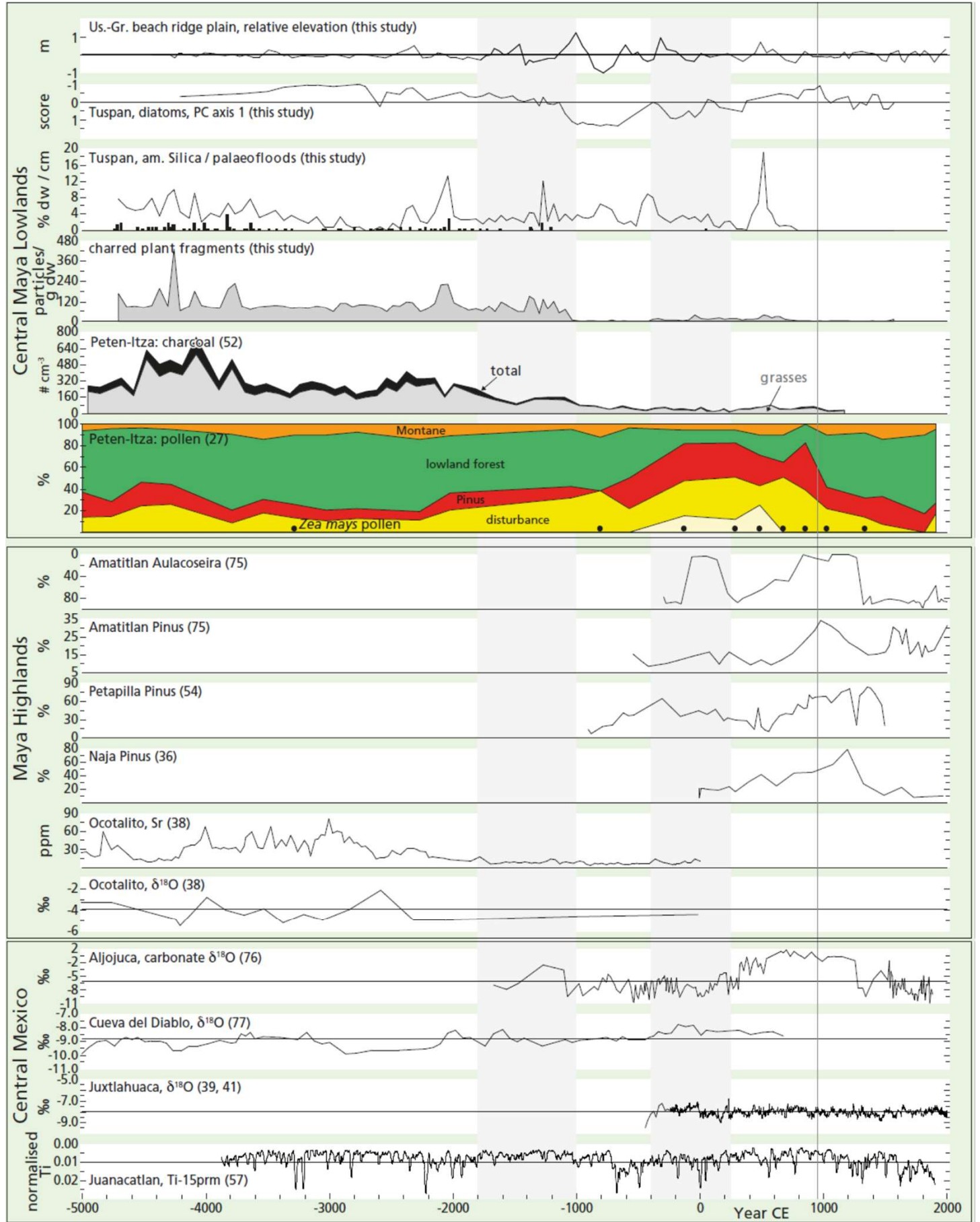

Fig. A2b

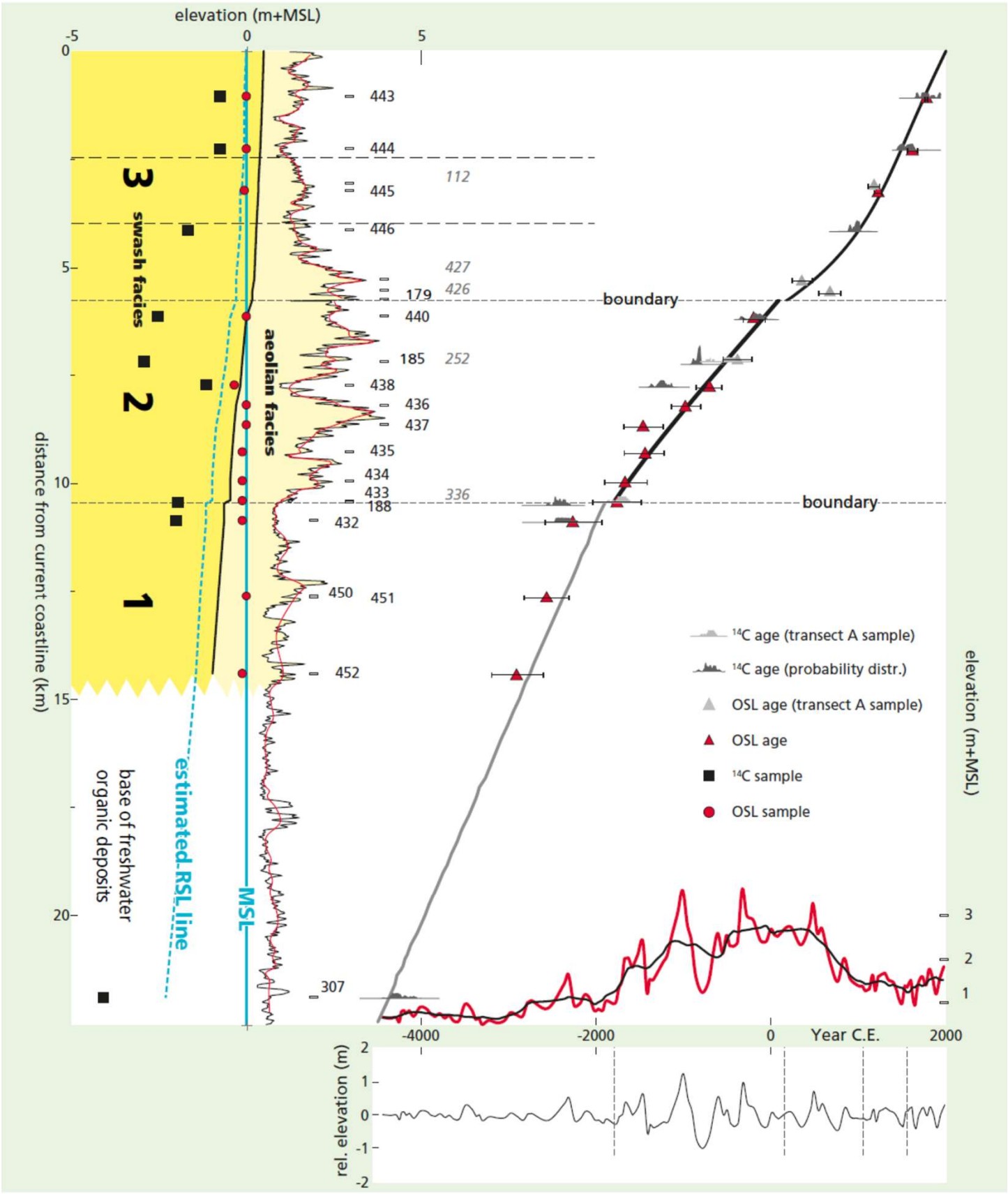

Fig. A3

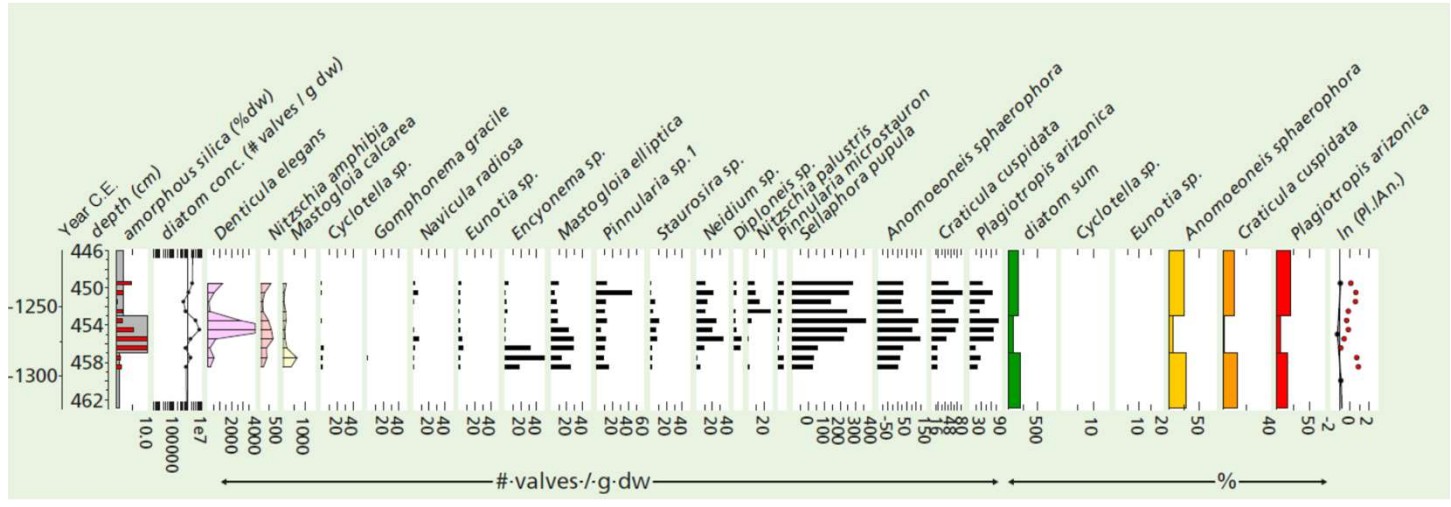

Fig. A4

Fig. A5

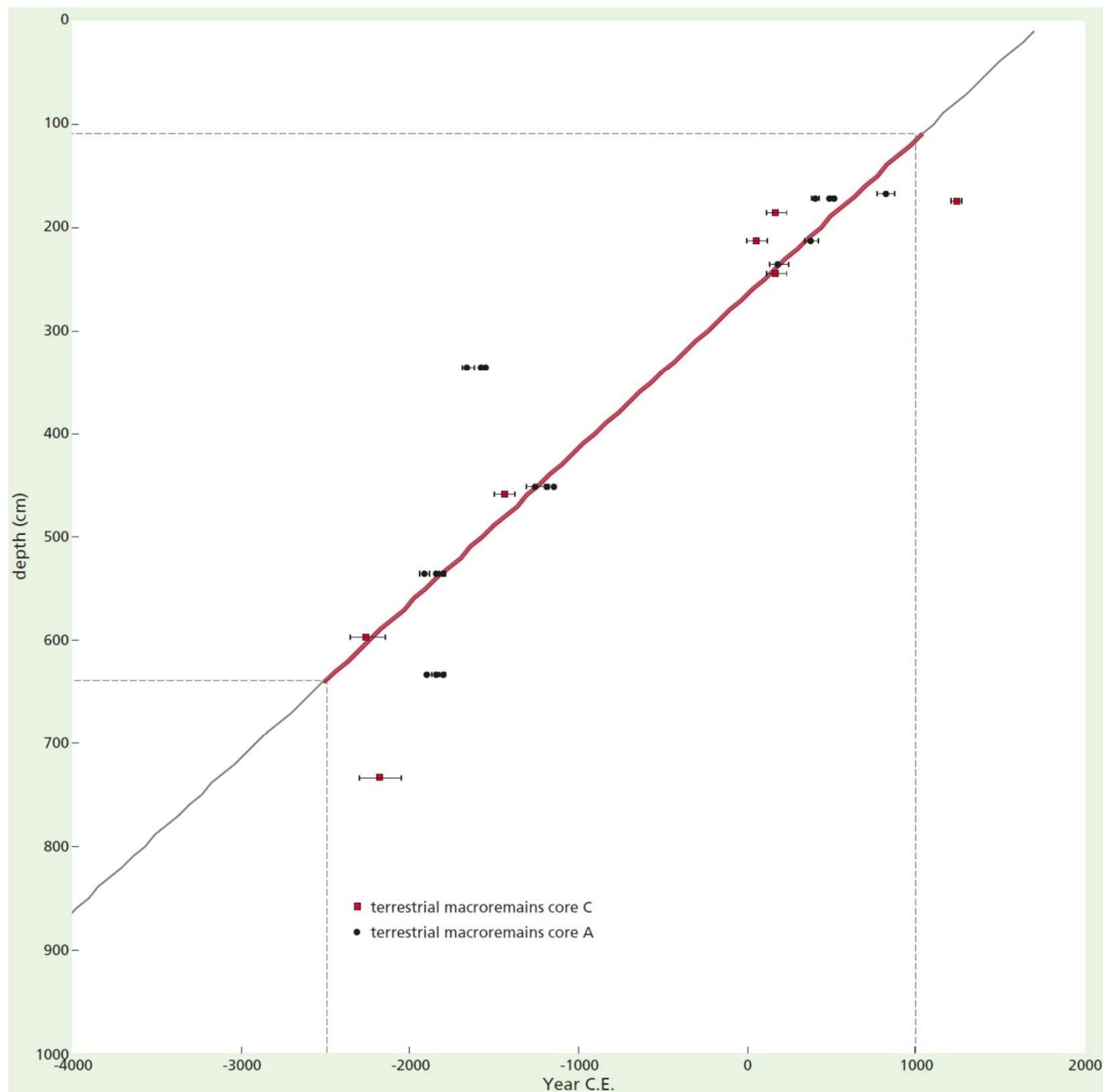

Fig. A6

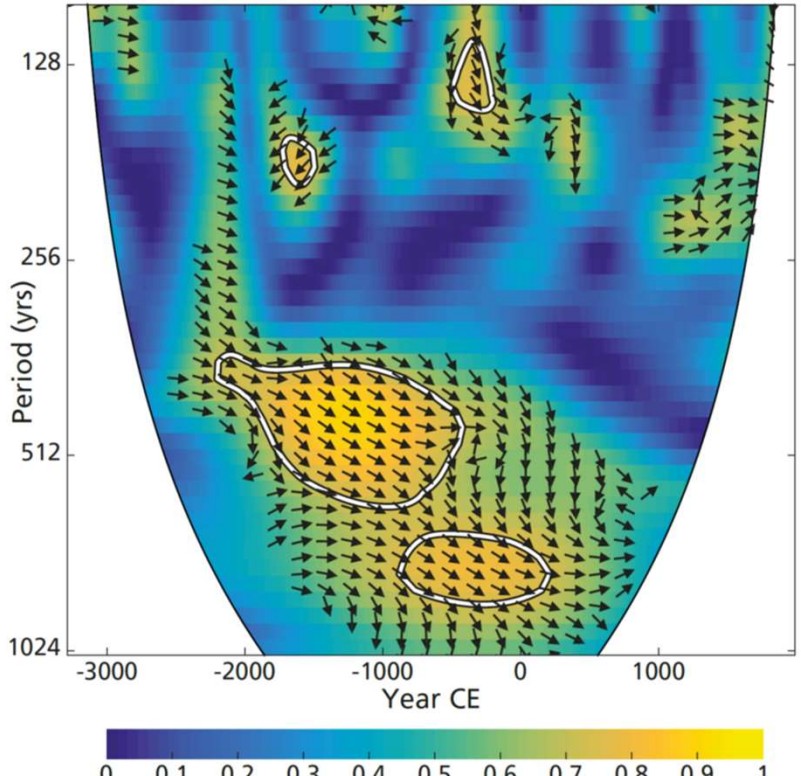

Fig. A7

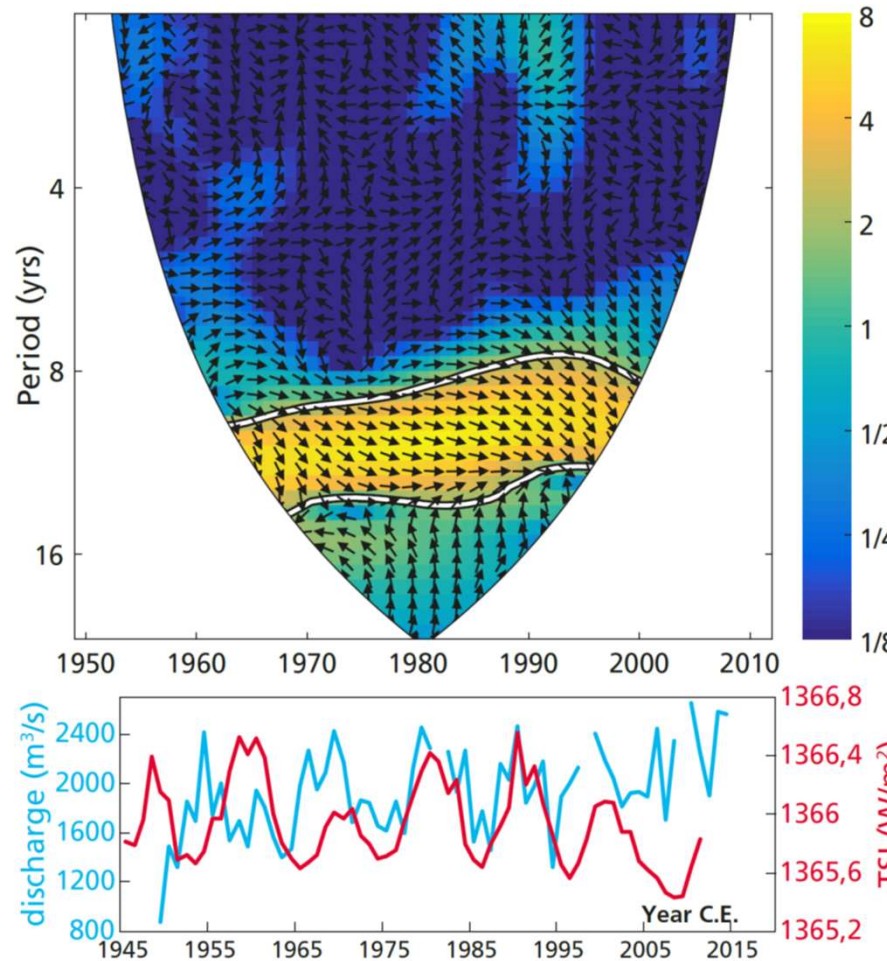

Fig. A8

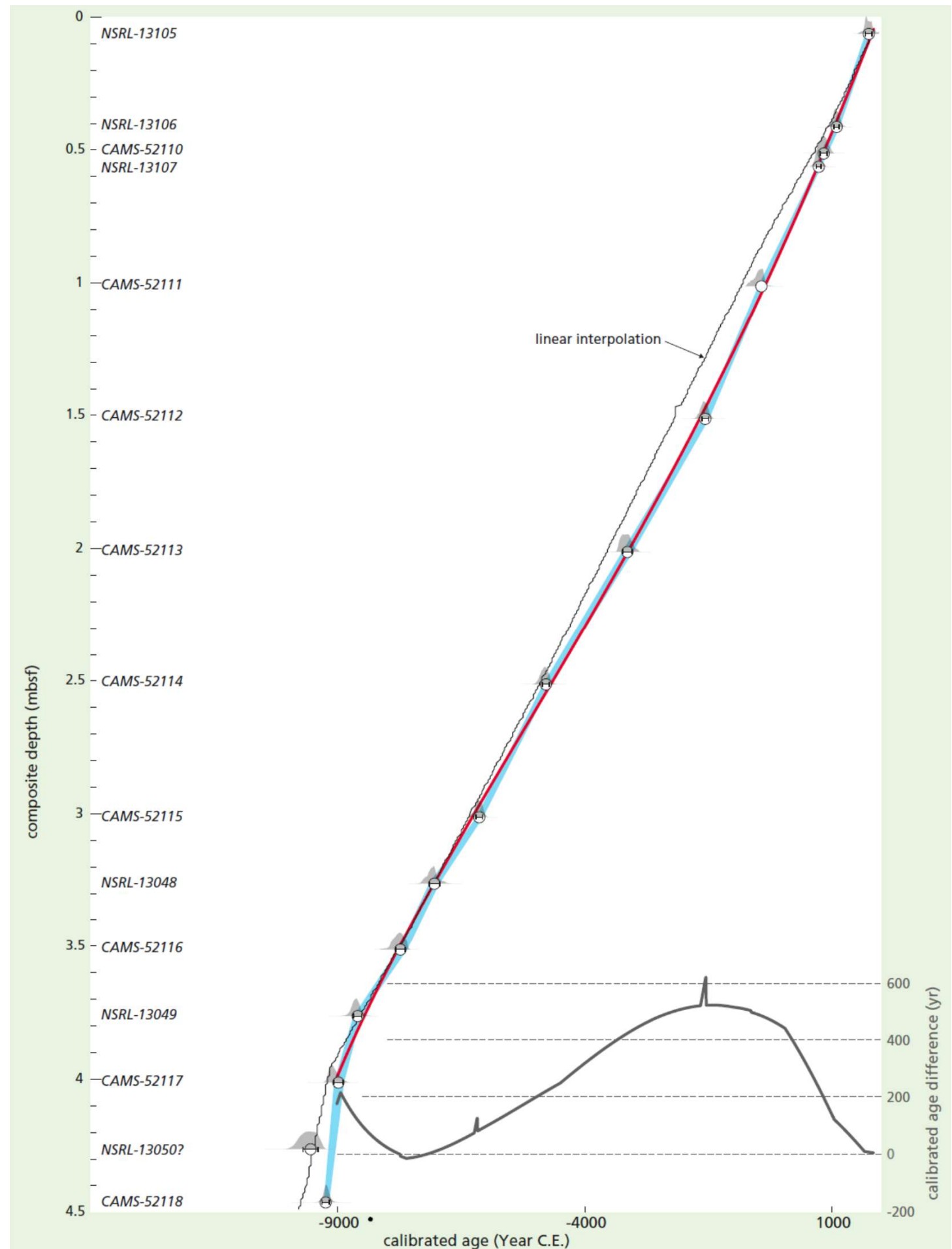

Fig. A9