# Peer review of "Climate impact on the development of Pre-Classic Maya civilization"

_Climate of the Past, 2018_

## Referee Comment (RC1) · Anonymous Referee #1 · 27 Mar 2018

Overall summary of comments: I found this paper to offer a new valuable source of paleodata for Mesoamerica, and believe that with some moderate revisions it will be an excellent paper for Climate of the Past. The paper is written well and clear, and it does an excellent job at investigating the wide range of existing paleorecords for the region. The beginning of the paper is particularly strong, but the structure and quality flags toward the end. Generally, more emphasis is needed on interpreting the data in light of the observations the authors point out.

The structure of the paper after the methods is quite muddled and difficult to determine a consistent focus. Reorganization of this section with clearly defined results separated from discussion of those results may help. For example, the portion from 232-281 feel very different than the earlier parts of section 3. The earliest parts read very much as

straight results, whereas the later parts attempt to discuss the drivers of these results. I feel these should be in different sections.

In the end, I believe that the paper brings some interesting ideas up backed by an excellent study and report of new paleodata. However, while the data itself and first half of the paper are excellent, the discussion of the results leads much to be desired. I believe many of main points of your discussion are already present in this manuscript, but the structure at present does not bring these points to the forefront. Many of these discussion points also need more development to describe how they relate to the existing body of knowledge, and how to interpret your new data with existing data in mind. Again, I think that the big conclusions exist in this paper, but they need to be written in a way that highlights them and supports them better.

Finally, the abrupt ending of the paper left me feeling like this was an incomplete draft. I actually thought that there would be at least another page of material based on the flow of the paper up to that point. Make sure the ideas you highlight at the beginning of the paper (e.g., role of floods in Maya myth, the benefit of having the paired but different size watersheds of the beach ridges and lakes, etc.) are brought up again or emphasized throughout the paper.

Line by line commentary 71-72: The use of 'likely' twice here reads a little awkward 106-111: Can you rephrase this sentence? It is very long and full of multiple clauses. I had to read it multiple times to make sense of it. I think splitting it up and simply restating it in another way would get your point across much better 131: In the methods section, you talk about elemental analysis through X-ray fluorescence and how you use that to identify floods, but in the background on Lake Tuspan, you only discuss diatoms. I was left trying to figure out why you brought up the elemental analysis and whether it was just supporting diatom conclusions or if you were using it as an independent proxy. Perhaps a few sentences in the background clarifying all the techniques you use to make a paleo precip lake signal would help. 152: Missing an extra line spacing here (not a big deal, just bringing it up for when you are doing final format

check) 167: In the background, you discuss the beach ridges before the lake study, but it's flipped in the methods. There's also some inconsistency in this order in the results/discussion. Again, this is not a critical problem, but readability can be enhanced if you keep the order of information similar between sections. 167: It would be nice to have a brief restatement (maybe in the background) on how these ridges were dated rather than having the reader look up cited literature. 186 (and prior): Your abstract makes it clear that this paper is reporting a paleorecord based on the combined record of the lake and the ridges. However, much of the background (e.g., line 92) emphasizes the beach ridge record and treats the lake as secondary/supplemental data. In section 3, the balance between the two records is much clearer, but later discussion again seems to emphasize the beach ridges. My takeaway impression at the end was that the lake data sometimes seemed haphazardly integrated into a study largely based on the beach ridge data. To help this, I don't think you really need more analysis or data from the lake; you just need to make sure that you are equally discussing the contributions of both in all sections and making it clear why you are favoring the lake or the ridges in some parts of the discussion. 223: There is decent evidence of regional drying at the close of the late Preclassic. Does your record support this? Or does it not extend far enough to be confident? 233: Long term drying trend in what? Your data? Perhaps describe what you think is the long-term trend of your data, because up to this point it is not clear based on your previous discussion (185-230) that there is a drying trend (it seems quite variable). 233-239: This whole sections reads more like background information to set up your study rather than discussing your findings. It also feels a bit out of place to discuss broad long term variability AFTER you've covered the short term, period by period results. 246: The Macal Chasm record shows broad similarities in the timing of long-period dry events with many of the other paleorecords in the region, although this is based on visual matching and not statistical work like wavelet analysis. Why do you think that the Macal Chasm record is the only one to show a match? Some discussion of whether you feel this is due to actual environmental differences or if it is data quality/characteristic driven would be nice here. Perhaps

the records that match well in the Classic and Postclassic show more divergence in the Preclassic? 251: The Macal Chasm chronology is not particularly precise, especially compared to other stalagmite records in the region like from Yok Balum. The uncertainties in the chronology run 200-400 years in many cases. Even if this doesn't adversely affect your conclusions, you may wish to address this and at the very least qualify how you decided that Macal Chasm is considered 'well-dated'. 269: Why would this period be different? If you are going to tell us that it isn't an analogue, you need to explain why this period is such an aberration. 241-281: I feel that this section is one of your weaker parts of the paper. I do not see a convincing argument laid out that the North Atlantic is driving your variations, although I catch a glimpse of it. I would start this section laying out how your record relates with the North Atlantic and atmospheric data and build the argument of what is driving the changes you observe in your data. I think that is the best and clearest argument you have in your whole paper, and you should highlight this. Then as a follow up, you can examine how your data relates to other regional records. This would be a nice set up for you to go into more discussion on why you think this North Atlantic signal expressed in your data isn't showing up in other data beside Macal Chasm. Your final paragraph on Cariaco is more what I am looking for in terms of discussing how/why your data has coherence/lack of coherence with other records. 284-305: I also feel that this section is underdeveloped. You are arguing that overly wet conditions may have delayed maize agriculture development, and I think this can be a valid hypothesis. However, you earlier pointed out that climate instability may be to blame for delayed maize (line 76). You also do not supply evidence for your 'overly wet' hypothesis in the form of maize physiology or ethnographic studies. If the region became wetter overall, some low lying areas would be too wet for agriculture, but wouldn't other regions that are presently too dry become potentially productive? Could it simply be a coincidence that local maize varieties hadn't been selected enough for local adaptation until the boom in agricultural clearance you note? Or that populations grew enough in the 'wet' years to support the increased social structures required for large scale agriculture and societal development, rather than

maize being actively suppressed by the climate? These alternatives may not be valid, but I don't feel that your argument for wet = bad for maize = suppression of societal development makes enough of a causative case to defend itself against alternative theories. In particular, many have argued that the Classic Period was relatively wetter (e.g., YOK-1, Chicancanab) and this drove societal development and population growth. Others (e.g., Macal Chasm) argued that their data doesn't support a wetter Classic and that other factors, such as climate stability, are more important than wetter vs drier. Where does your research land on this issue? Overall, I think you need to discuss the Maya-environment interactions in much more depth if you wish to make arguments of a somewhat environmentally-deterministic nature. 306: A very abrupt end without any concluding statements. I was left with, "Wait, what was the point or points they really wanted me to focus on?"

---

## Referee Comment (RC2) · M. Brenner (Referee) · 28 Apr 2018

Nooren et al. argue that development of ancient Maya Civilization, much as has been argued for its later disintegration, was highly dependent on climate conditions. Specifically, the authors attribute early agricultural expansion and urban development to drying in the Preclassic. The conclusions come from study of diatoms in a sediment core from Lake Tuspan, and changes in beach ridge elevation near the Usumacinta discharge on the Gulf of Mexico coast. The authors note that it has been difficult to tie all paleoclimate records from the Maya Lowlands together, and that there are apparent disagreements between climate "reconstructions." This may reflect, in part, problems with dating such records. But the authors do an admirable job of pulling together numerous records (paleolimnological, paleoceanographic, and speleothem). The paper

should be published and will be an important resource for archaeologists, ecologists and paleoclimatologists working in the region. I concur that late Holocene drying correlates temporally with the rise of Maya civilization, and have considered whether climate conditions finally became optimal for slash-and-burn (SAB) agriculture, i.e. just the right seasonality to fell the forest, dry it out, burn it and plant. I still have trouble sorting out how climate, in terms of total rainfall, might have prevented such activity earlier, give the dramatic gradient in annual rainfall across the Yucatan Peninsula today (especially north-south), which does not appear to prevent people from practicing SAB. Again, perhaps it had something to do with both how much rain fell and how it was distributed throughout the year. My comments are largely of an editorial nature and I will simply go through the manuscript and comment on a line-to-line basis.

Line

26 I'd change "while" to "whereas" as you are contrasting what the two records show 28 I believe it is more conventional to use a little pyramid symbol for the delta 14C 47 I suggest something like "Evidence for such impacts is found in the fact that floods, as well as droughts, are important themes. . . 49 change "Mayan" to "Maya" (the former refers to the language(s)) 58 Tankersley et al. also made the pitch that the "Maya clay" had a volcanic origin. But it is important to be clear about how that is meant. It may well be true that much of the smectite clays come from weathering of ash, deposited over very long periods. But I think Tankersley et al. were referring to the massive deposits, e.g. in Peten lakes, having come from Holocene eruptions. There are many good arguments against that, like the fact that the clay is meters thick in some of the lakes, and the fact that even in lakes surrounded by active volcanoes, one sees ash layers, intercalated with organic lake sediment (Amatitlan). My guess is that land clearance on steep slopes in Peten, under relatively wet conditions (compared to the deglacial, e.g.), enabled rapid export of fine particles from local soils. That is a little off topic. 72 delete "likely" – it appears on line 71 85-86 It may be that many of the differences among paleo-precipitation records reflect constraints on dating. The

only reliable lake-sediment dates come from terrestrial fossils, and we are still stuck with interpolating/extrapolating dates for depths where there are no datable materials. 97 change "70,700 km2 large catchment" to "70,700-km2 catchment" 108 change "has primarily been determined" to "was primarily determined" 136 Maybe it is worth noting that this is not the Rio Dulce that drains Lake Izabal (eastern Guatemala) 140 change to "exceeding a one-standard-deviation threshold" 164 change "by" to "of" 179 change "hematite stained" to "hematite-stained" 229 change "for" to "of" 233 change "last thousands of years" to "last few thousand years" 241 change "Centennial scale" to "Centennial-scale" 245 You use "The in-phase relationship between the two records is significant above a 5% confidence level at centennial timescales during the Pre-Classic Period." As stated, I am not sure what that means. Do you really mean that you set the alpha value at 5%, and the probability of concluding the records are in-phase, when in fact they are not, is <5%. I think that should be re-worded for clarity. 248 change to "at a centennial time scale" 252 change to "gives us confidence" (refers back to "The coherence") 254 I believe it is more conventional to use a little pyramid symbol for the delta 14C 257 I believe it is more conventional to use a little pyramid symbol for the delta 14C 258 change to "∼500-year" 266 change to "At that time" 276 change to "centennial-scale" 279 change "due to" to "as a consequence of" and later in the line to "because of" 284 change to "During that period" 290 change to "Between 1000 and 850 BCE" 292 change "at" to "on" 295 change "for further development" 301 change to "show strong and steady development" 304-305 Again, I wonder if it is drier conditions, or perhaps as important, how the rainfall was distributed through the year. Agriculture is practiced across a large gradient of annual rainfall today, using traditional methods. 623 change extend" to "extent" or "area" 630 I did not know what was meant by "the Cariaco record is conform updated age-depth model." Why not just say "We used an updated age-depth model for the Cariaco record." 632 I believe it is more conventional to use a little pyramid symbol for the delta 14C 637 I believe it is more conventional to use a little pyramid symbol for the delta 14C 639 change to "500-yr" 653 insert a period after "et al" 653 change to "Long-term" 668 Italicize "Aulacoseira" and "Pinus" (the latter

in 3 places) 671 change "Ti -15 point running mean" to "Ti 15-point running mean" 675 change to "1-4-cm-thick" 676 change to "light-coloured" 685 change to "concentrations" 687 change to "4-12-cm" 690 change to "events" 692 change to "linear" 693 change to "radiocarbon-dated" 704 change "at al." to "et al." 711 change to "linear" 711 change to "4th-order"

Figure A1. How was it decided which archaeological sites to include? There are certainly many more, and this may mislead readers who are unfamiliar with the archaeology of the region. Also, might another colour be used for the Dulce River catchment. It appears that the area received >4000 mm/yr rainfall, being dark blue.

Figures 3, A2a, A2b. Is there any utility in indicating on those plots which way is drier and which is wetter? Also, for A2b, I suspect that the orange pollen percentages for Peten-Itza are "Montane" rather than "Montana" Figure A10 – change to "linear"

---

## Author Comment (AC1) · 13 Jun 2018

Reply on "Climate impact on the development of Pre-Classic Maya civilization" (RC-1)

We thank the two reviewers for their constructive comments on our paper "Climate impact on the development of Pre-Classic Maya civilization". We are happy to read that according to the reviewers our two palaeo-precipitation records add valuable new sources of palaeodata for the understanding of human environmental interaction in the Central Maya Lowlands (RC-1 and RC-2).

Hereby we would like to reply on the comments of RC-1.

1. General commentary (RC-1)

Many of the main comments of RC-1 are related to the structure of the paper:

-The structure of the paper after the methods is quite muddled and difficult to determine a consistent focus. Reorganization of this section with clearly defined results separated from discussion of those results may help.

-I believe many of main points of your discussion are already present in this manuscript, but the structure at present does not bring these points to the forefront.

-Finally, the abrupt ending of the paper left me feeling like this was an incomplete draft.

-Make sure the ideas you highlight at the beginning of the paper (e.g., role of floods in Maya myth, the benefit of having the paired but different size watersheds of the beach ridges and lakes, etc.) are brought up again or emphasized throughout the paper.

We followed RC-1 recommendations and thoroughly restructured paragraph 3 and 4, and added a paragraph with conclusions which will hopefully meet RC-1 expectations.

3. Results

Beach-ridge record

Diatom record lake Tuspan

Wavelet transfer functions

4. Discussion

4.1 Climate change in the CML during the Pre-Classic period

Early Pre-Classic period (1800 – 1000 BCE)

Middle Pre-Classic period (1000 – 400 BCE)

Late Pre-Classic period (400 BCE – 250 CE)

4.2 Precipitation variability

Precipitation variability over long time scales

Centennial scale precipitation variability

4.3 Precipitation versus human development in the CML

5. Conclusions

5. Conclusions

For the first time a regional palaeo-precipitation record has been reconstructed for the Central Maya Lowland (CML), based on an exceptionally well dated high resolution beach ridge record. This record indicates centennial scale precipitation fluctuations during the Pre-Classic period that are not always registered in local records, adding valuable new insights into larger scale climatic forcing mechanisms for the CML. The generally poor correlation between the regional and local palaeo-precipitation reconstructions are probably related to spatial precipitation variability, and chronological uncertainties of many records. Additional research of beach ridge formation processes are needed to extend this regional precipitation reconstruction to the Classic and Post-Classic period. We have also generated a local scale palaeo-precipitation record using diatoms preserved in a core from Lake Tuspan, thereby adding an alternative proxy to the relatively high number of local reconstructions predominantly based on oxygen isotope variability. We recognise, however, that diatom preservation is often poor in the carbonate lakes across the wider region. As a result, the correlation between these two reconstructions is variable through time.

Although the occurrence of a prolonged drought during the end of the Early Pre-Classic period, which we report here, is evident in other palaeo-precipitation reconstructions from the CML, the subsequent wet period during the Middle Pre-Classic period, registered in both our new records, is less evident elsewhere. Although many researchers have focused on the impact of drought on the development and disintegration of Maya societies, one should consider this prolonged wet period as potentially unfavourable for

the development and intensification of agriculture in the CML, particularly in the wetter areas. We cannot be certain about the impact of wetter conditions on the Maya. However, owing to the lack of the development at this time we theorise that the wet period could have created poor growing conditions for maize in the CML. In order to test theory, we advocate for the use of process-based modelling approaches which capture heterogeneous environmental constraints on crop growth for given climate boundary conditions such as the approach applied by Dermody et al. (2014).

Our results provide evidence that North Atlantic atmospheric-oceanic forcing plays an important role in the modulation of the observed centennial scale precipitation variability, however further studies are required which compare well-dated terrestrial reconstructions that capture regional signals with solar and oceanic reconstructions to gain a better understanding of climate forcing mechanisms, both in the CML and across the wider region.

Dermody, B.J., van Beek, R.P.H., Meeks, E., Klein Goldewijk, K., Scheidel, W., van der Velde, Y., Bierkens, M.F.P., Wassen, M.J., Dekker, S.C., A virtual water network of the Roman world. Hydrol. Earth Syst. Sci. 18, 5025–5040, 2014.

2. Line by line commentary (RC-1)

71-72: The use of 'likely' twice here reads a little awkward

We agree: the second 'likely' has been removed.

106-111: Can you rephrase this sentence? It is very long and full of multiple clauses.

We agree. Sentence is split into two sentences: Although multiple factors determine the final elevation of the beach ridges, it has been shown that during the period 1775 $\pm$ 95 BCE to 30 $\pm$ 95 CE (at 1Æą), roughly coinciding with the Pre-Classic period, beach ridge elevation has primarily been determined by the discharge of the Usumacinta river. Low elevation anomalies of the beach ridges occur in periods with increased river sediment discharge, which in turn is the product of high precipitation within the

river catchment.

131: In the methods section, you talk about elemental analysis through X-ray fluorescence and how you use that to identify floods, but in the background on Lake Tuspan, you only discuss diatoms. I was left trying to figure out why you brought up the elemental analysis and whether it was just supporting diatom conclusions or if you were using it as an independent proxy. Perhaps a few sentences in the background clarifying all the techniques you use to make a paleo precip lake signal would help.

We agree. We will give a more thorough description of lake Tuspan's core lithology and XRF results in the Results section. Appendix figure A4 will therefore be moved to the main text. This will likely also accommodate RC-1 request for a more balanced presentation of both records [much of the background (e.g., line 92) emphasizes the beach ridge record and treats the lake as secondary/supplemental data].

152: Missing an extra line spacing. Line spacing added.

167: In the background, you discuss the beach ridges before the lake study, but it's flipped in the methods.

Indeed, we corrected this, and in the discussion section we more consistently discuss the beach ridge record first, followed by the diatom record.

167: It would be nice to have a brief restatement (maybe in the background) on how these ridges were dated rather than having the reader look up cited literature.

We agree. We therefore added the following sentence to the text: The age distance model is based on 35 AMS 14C dated terrestrial macro-remains (mainly leaf fragments isolated from organic debris layers), and 20 OSL dated sand samples (determined on small aliquots of quartz grains) (Nooren et al., 2017b).

223: There is decent evidence of regional drying at the close of the late Preclassic. Does your record support this? Or does it not extend far enough to be confident?

We found a pronounced dry interval during the early part of the Late Pre-Classic Period centred around 275 BCE, but a drought around the close of the late Pre-Classic Period (around 150 – 250 CE), often related to the Pre-Classic collapse, occurred just after the period covered by our beach ridge record (1775 ± 95 BCE to 30 ± 95 CE (at 1Æą)).

233: Long term drying trend in what? Your data? Perhaps describe what you think is the long-term trend of your data, because up to this point it is not clear based on your previous discussion (185-230) that there is a drying trend (it seems quite variable).

We agree. We moved the sentence to the Result section (Diatom record lake Tuspan), and rewrote it as: After relatively high/positive PC-1 values during the Middle and Late Pre-Classic Period we observe a decreasing trend in PC-1 values during the following Classic Period, indicating a gradual increase in lake water salinity. Low PC-1 values between 800 – 950 CE are in accordance with many palaeorecords from the area (Fig. A2) indicating periods of prolonged droughts during the Late Classic Period.

233-239: This whole sections reads more like background information to set up your study rather than discussing your findings. It also feels a bit out of place to discuss broad long term variability AFTER you've covered the short term, period by period results.

We moved this section to paragraph 4.2 (Precipitation variability) of the discussion section. We think that our given explanation of the long term drying trend is more than background information, and should be presented here.

RC-1: The Macal Chasm record shows broad similarities in the timing of long-period dry events with many of the other paleorecords in the region, although this is based on visual matching and not statistical work like wavelet analysis. Why do you think that the Macal Chasm record is the only one to show a match? Some discussion of whether you feel this is due to actual environmental differences or if it is data quality/characteristic driven would be nice here.

We only performed our wavelet analyses between the beach ridge record and all proxy records presented in Fig. A3. Therefore, we are not able to quantitatively compare the Macal Chasm record with other local records. The coherence between the beach ridge record and the Macal-Chasm record may be related to the fact that both records are relatively well dated (as stated in line 251).

RC-1: The Macal Chasm chronology is not particularly precise, especially compared to other stalagmite records in the region like from Yok Balum. The uncertainties in the chronology run 200-400 years in many cases. Even if this doesn't adversely affect your conclusions, you may wish to address this and at the very least qualify how you decided that Macal Chasm is considered 'well-dated'.

We will add 'relatively', and describe the Macal Chasm chronology as relatively well-dated. Although chronological uncertainties of the Macal Chasm record during the Pre-Classic Period are generally in the order of 150 years (at 1 Æą), this is better than most other palaeorecords for this time period. The Yok Balum record encompassed the Classic and Post-Classic period but hardly extend into the Pre-Classic.

269: Why would this period be different? If you are going to tell us that it isn't an analogue, you need to explain why this period is such an aberration.

Good question, we actually don't fully understand how climate forcing mechanisms during the Pre-Classic Period were different from today. We will add after line 271: Probably due a more northerly mean position of the ITCZ during the Pre-Classic period precipitation responded differently to solar forcing then today.

241-281: I feel that this section is one of your weaker parts of the paper. I do not see a convincing argument laid out that the North Atlantic is driving your variations, although I catch a glimpse of it. I would start this section laying out how your record relates with the North Atlantic and atmospheric data and build the argument of what is driving the changes you observe in your data.

We agree that that the centennial scale precipitation variability is likely driven by North Atlantic atmospheric-oceanic forcing forms an important finding of our research. However further research, and the input of climatologists are needed to understand the observed correlation and climate forcing mechanisms. We have added this to the conclusions.

284-305: I also feel that this section is underdeveloped. You are arguing that overly wet conditions may have delayed maize agriculture development, and I think this can be a valid hypothesis. However, you earlier pointed out that climate instability may be to blame for delayed maize (line 76). You also do not supply evidence for your 'overly wet' hypothesis in the form of maize physiology or ethnographic studies. If the region became wetter overall, some low lying areas would be too wet for agriculture, but wouldn't other regions that are presently too dry become potentially productive? Could it simply be a coincidence that local maize varieties hadn't been selected enough for local adaptation until the boom in agricultural clearance you note? Or that populations grew enough in the 'wet' years to support the increased social structures required for large scale agriculture and societal development, rather than maize being actively suppressed by the climate? These alternatives may not be valid, but I don't feel that your argument for wet = bad for maize = suppression of societal development makes enough of a causative case to defend itself against alternative theories. In particular, many have argued that the Classic Period was relatively wetter (e.g., YOK-1, Chicancanab) and this drove societal development and population growth. Others (e.g., Macal Chasm) argued that their data doesn't support a wetter Classic and that other factors, such as climate stability, are more important than wetter vs drier. Where does your research land on this issue? Overall, I think you need to discuss the Maya-environment interactions in much more depth if you wish to make arguments of a somewhat environmentally-deterministic nature.

We are glad that reviewer 1 considers our hypothesis that wet conditions may have delayed the development of maize agriculture a valid hypothesis. However his suggestion for a more in depth discussion of the agricultural and archaeological implications goes beyond the scope of this article, and will need further research, particularly from an archaeological point of view. At this stage we think it is important to point out that one should not only consider prolonged droughts as negative for the development of human societies in the area. We have added in the main text the following: We cannot be certain about the impact of wetter conditions on the Maya. However, owing to the lack of the development at this time we theorise that the wet period could have created poor growing conditions for maize in the CML. In order to test theory, we advocate for the use of process-based modelling approaches which capture heterogeneous environmental constraints on crop growth for given climate boundary conditions such as the approach applied by Dermody et al. (2014).

306: A very abrupt end without any concluding statements. I was left with, "Wait, what was the point or points they really wanted me to focus on?"

A paragraph with conclusions has been added to the text.

---

## Author Comment (AC2) · 13 Jun 2018

Reply on "Climate impact on the development of Pre-Classic Maya civilization" (RC-2)

We thank the two reviewers for their constructive comments on our paper "Climate impact on the development of Pre-Classic Maya civilization". We are happy to read that according to the reviewers our two palaeo-precipitation records add valuable new sources of palaeodata for the understanding of human environmental interaction in the Central Maya Lowlands (RC-1 and RC-2).

Hereby we would like to reply on the comments of RC-2.

1. Line by line commentary (RC-2)

[Figure]

I'd change "while" to "whereas" as you are contrasting what the two records show

Agree.

I believe it is more conventional to use a little pyramid symbol for the delta 14C

Both are in use.

I suggest something like "Evidence for such impacts is found in the fact that floods, as well as droughts, are important themes:

Agree.

change "Mayan" to "Maya" (the former refers to the language(s))

Agree

Tankersley et al. also made the pitch that the "Maya clay" had a volcanic origin. But it is important to be clear about how that is meant.

Agree, we have added a reference to Tankersley et al. (2016) and changed sentence 58: ...., and past volcanic activity could have been responsible for the deposition of smectite rich clay layers in inland lakes (Tankersley et al., 2016; Nooren et al., 2017a).

Tankersley, K.B., Dunning, N.P., Scarborough, V., Huff, W.D., Lentz, D.L., and Carr, Catastrophic volcanism and its implication for agriculture in the Maya Lowlands: Journal of Archaeological Science 5, 465–470, 2016.

delete "likely" – it appears on line 71

Agree.

85-86 It may be that many of the differences among paleo-precipitation records reflect constraints on dating.

Indeed, few well-dated records exist for the Central Maya Lowlands.

change "has primarily been determined" to "was primarily determined". Done.

Maybe it is worth noting that this is not the Rio Dulce that drains Lake Izabal (eastern Guatemala). Done.

change to "exceeding a one-standard-deviation threshold". Done.

change "by" to "of". Done.

change "hematite stained" to "hematite-stained". Done.

change "for" to "of". Done.

change "last thousands of years" to "last few thousand years". Done.

change "Centennial scale" to "Centennial-scale". Done.

You use "The in-phase relationship between the two records is significant above a 5% confidence level at centennial timescales during the Pre-Classic Period." As stated, I am not sure what that means. Do you really mean that you set the alpha value at 5%, and the probability of concluding the records are in-phase, when in fact they are not, is <5%. I think that should be re-worded for clarity.

We have added the following to section 2 (Methods): CWT applies Monte Carlo methods to test for significance. In this case we set the alpha value at 5%. Time periods and periodicities enclosed within the black lines of in our wavelet analysis indicate common power between timeseries with 95% confidence.

change to "at a centennial time scale". Done.

change to "gives us confidence" (refers back to "The coherence"). Done.

and 257 I believe it is more conventional to use a little pyramid symbol for the delta 14C. Both are in use.

change to "Âň500-year". Done.

change to "At that time". Done.

change to "centennial-scale". Done.

change "due to" to "as a consequence of" and later in the line to "because of". Done.

change to "During that period". Done.

change to "Between 1000 and 850 BCE". Done.

change "at" to "on". Done.

change to "for further development". Done.

change to "show strong and steady development". Done.

304-305 Again, I wonder if it is drier conditions, or perhaps as important, how the rainfall was distributed through the year. Agriculture is practiced across a large gradient of annual rainfall today, using traditional methods.

We agree. We added after sentence 305: Changes in the distribution of rainfall probably also changed, and large floods, most evident during the Archaic and early Pre-Classic period, occurred much less frequently after approximately 1000 BCE.

change extend" to "extent" or "area". Done.

I did not know what was meant by "the Cariaco record is conform updated age-depth model." Why not just say "We used an updated age-depth model for the Cariaco record." Done.

and 637 I believe it is more conventional to use a little pyramid symbol for the delta 14C. Both are in use.

change to "500-yr". Done.

insert a period after "et al". Done.

change to "Long-term". Done.

Italicize "Aulacoseira" and "Pinus" (the latter in 3 places). Done.

change "Ti -15 point running mean" to "Ti 15-point running mean". Done.

change to "1-4-cm-thick". Done.

change to "light-coloured". Done.

change to "concentrations". Done.

change to "4-12-cm". Done.

change to "events". Done.

change to "linear". Done.

change to "radiocarbon-dated". Done.

change "at al." to "et al.". Done.

change to "linear". Done.

change to "4th-order" Done.

Figure A1. How was it decided which archaeological sites to include? There are certainly many more, and this may mislead readers who are unfamiliar with the archaeology of the region. Also, might another colour be used for the Dulce River catchment. It appears that the area received >4000 mm/yr rainfall, being dark blue.

The colour of the Dulce river catchment has been adjusted, and in the figure legend 'archaeological site' has been changed to 'major archaeological site'.

Figures 3, A2a, A2b. Is there any utility in indicating on those plots which way is drier and which is wetter? Also, for A2b, I suspect that the orange pollen percentages for Peten-Itza are "Montane" rather than "Montana"

We have added some arrows in Figure 4 (was Fig. 3) to indicate if excursion indicate drier of wetter conditions. We added to Figure caption A2: Notice that the y-axis is sometimes reversed, so that excursion above the x-axis always indicate relatively drier conditions.

Montane indeed !

Figure A10 (now Fig. A9) – change to "linear". Done.
* * *